# High-performance blue OLED using multiresonance thermally activated delayed fluorescence host materials containing silicon atoms

Dongmin Park[1,6], Seokwoo Kang[2,6], Chi Hyun Ryoo[1], Byung Hak Jhun[1], Seyoung Jung[1], Thi Na Le[3], Min Chul Suh[3], Jaehyun Lee[4], Mi Eun Jun[5], Changwoong Chu[5], Jongwook Park[2] ✉ & Soo Young Park[1] ✉

We report three highly efficient multiresonance thermally activated delayed fluorescence blue-emitter host materials that include 5,9-dioxa-13b-boranaphtho[3,2,1-*de*]anthracene (DOBNA) and tetraphenylsilyl groups. The host materials doped with the conventional $N^7,N^7,N^{13},N^{13},5,9,11,15$-octaphenyl-5,9,11,15-tetrahydro-5,9,11,15-tetraaza-19b,20b-diboradinaphtho[3,2,1-*de*:1',2',3'-*jk*]pentacene-7,13-diamine (*v*-DABNA) blue emitter exhibit a high photoluminescence quantum yield greater than 0.82, a high horizontal orientation greater than 88%, and a short photoluminescence decay time of 0.96–1.93 µs. Among devices fabricated using six synthesized compounds, the device with (4-(2,12-di-*tert*-butyl-5,9-dioxa-13*b*-boranaphtho[3,2,1-*de*]anthracen-7-yl)phenyl)triphenylsilane (TDBA-Si) shows high external quantum efficiency values of 36.2/35.0/31.3% at maximum luminance/500 cd m$^{-2}$/1,000 cd m$^{-2}$. This high performance is attributed to fast energy transfer from the host to the dopant. Other factors possibly contributing to the high performance are a $T_1$ excited-state contribution, inhibition of aggregation by the bulky tetraphenylsilyl groups, high horizontal orientation, and high thermal stability. We achieve a high efficiency greater than 30% and a small roll-off value of 4.9% at 1,000 cd m$^{-2}$ using the TDBA-Si host material.

Organic light-emitting diodes (OLEDs) have been commercialized using phosphors with internal quantum efficiencies (IQEs) of 100% in the green and red emission regions[1]. However, in the case of blue phosphorescent materials, the efficiency of OLEDs is low and their lifetime is short because of high bandgap energy and poor stability. The introduction of thermally activated delayed fluorescence (TADF)

by Adachi and coworkers has led to the development of materials with IQEs as high as 100% through reverse intersystem crossing (RISC)[2]. However, these materials have wide full-widths at half-maximum (FWHMs) of 60–100 nm and exhibit low color purity. To overcome this problem, Hatakeyama and coworkers introduced the concept of multiresonance TADF (MR-TADF), which reduces the $\Delta E_{ST}$ by localizing

[1]Center for Supramolecular Optoelectronic Materials (CSOM), Department of Materials Science and Engineering, Seoul National University, 1 Gwanak-ro, Gwanak-gu, Seoul 08826, Republic of Korea. [2]Integrated Engineering, Department of Chemical Engineering, Kyung Hee University, Gyeonggi 17104, Republic of Korea. [3]Department of Information Display, Kyung Hee University, Dongdaemoon-Gu, Seoul 02447, Republic of Korea. [4]Advanced Chemical Materials R&D Team, Korea Testing & Research Institute, Gwangyang 57765, Republic of Korea. [5]Samsung Display, 1 Samsung-ro Giheung-Gu, Yongin 17113, Republic of Korea. [6]These authors contributed equally: Dongmin Park, Seokwoo Kang. ✉e-mail: jongpark@khu.ac.kr; parksy@snu.ac.kr

the orbital density of the highest occupied molecular orbital (HOMO) and lowest unoccupied molecular orbital (LUMO) at different atoms. MR-TADF materials have shown a narrow FWHM of ~20 nm and a high photoluminescence quantum yield (PLQY) in excess of 90%[3–5].

Among the blue-emitting MR-TADF dopants reported to date, $N^7,N^7,N^{13},N^{13},5,9,11,15$-octaphenyl-5,9,11,15-tetrahydro-5,9,11,15-tetraaza-19b,20b-diboradinaphtho[3,2,1-de:1',2',3'-jk]pentacene-7,13-diamine (v-DABNA) exhibits the highest external quantum efficiency (EQE). Specifically, it exhibited an EQE of 34.4% when used as a dopant at a concentration of 1%[6,7]. For v-DABNA to be an effective dopant, energy transfer from the host to the dopant must be effective and aggregation of v-DABNA must be avoided, both of which strongly depend on the host material. However, despite numerous studies on MR-TADF dopants, until recently only a few studies have examined host materials other than 5,9-dioxa-13b-boranaphtho[3,2,1-de]anthracene (DOBNA)[8–10], which has been frequently used as a basic core moiety of host materials because of its TADF characteristics and high triplet energy[11].

An additional issue with conventional MR-TADF devices is that they exhibit severe efficiency roll-off behavior, which is related to the correlation between the properties of the host material and those of the dopant. This behavior is related not only to the decrease of the triplet-state exciton but also to the effective and fast energy transfer from host to dopant[12,13]. Specifically, a strong electric field creates excess triplet excitons in the host and these must be transferred quickly to the dopant. The objective of the present study was to develop a host material capable of achieving a high efficiency of 30% even at 1000 cd m$^{-2}$.

In the present study, to develop a host material to be used in combination with the blue MR-TADF dopant, we synthesized six host materials using 2,12-di-tert-butyl-5,9-dioxa-13b-boranaphtho[3,2,1-de] anthracene (TDBA), a derivative of the DOBNA moiety with MR-TADF. In these materials, to inhibit intermolecular interactions, we substituted either phenyl groups (TDBA-Ph, mTDBA-Ph, and mTDBA-2Ph) or tetraphenylsilane (TPS) groups (TDBA-Si, mTDBA-Si, and mTDBA-2Si) at the para- and meta- positions relative to the boron atom of the

TDBA moiety[14–16]. The latter three materials are expected to be suitable host materials because they all exhibit TADF and exhibit enhanced thermal stability and film-formation behavior. The structural design concept of the host material for our proposed MR-TADF emitter is the inclusion of silicon atoms. These atoms have the function of increasing the intermolecular distance in the host material.

To confirm the performance of the host materials, we fabricated devices in which the host material was doped with v-DABNA. The spin-orbit coupling (SOC) of the $S_1$ and $T_1$ states and of the $S_1$ and $T_2$ states of the materials were calculated. Energy transfer from the host materials to the dopant was confirmed to occur via Förster resonance energy transfer (FRET) through the $T_1$ state to the singlet states. As a result, non-radiative processes such as triplet–triplet annihilation (TTA) and triplet–polaron quenching (TPQ) can be reduced. Because of this effect, a blue OLED device based on TDBA-Si was developed that exhibited not only an ultra-high EQE$_{max}$ of 35.0% but also a small roll-off of less than 4.9% at a high luminance of 1000 cd m$^{-2}$.

## Results
### Molecular design concept and synthesis

Figure 1a shows the molecular structures of the synthesized host materials. These host materials have a DOBNA-type main backbone that exhibits the MR-TADF effect. We developed six host materials in which one or more phenyl or TPS groups were substituted onto the DOBNA-type main backbone, i.e., TDBA-Ph, mTDBA-Ph, mTDBA-2Ph, TDBA-Si, mTDBA-Si, and mTDBA-2Si (Fig. 1a). The TPS moiety is a bulky structure with a long C−Si bond length; it can therefore effectively prevent packing of molecules in the solid state. Single crystal analysis of TDBA-Ph and TDBA-Si confirmed that TDBA-Si containing the TPS moiety had a longer plane-to-plane distance than did TDBA-Ph (Supplementary Fig. S1). The presence of TPS moieties thus enables the formation of amorphous thin films and leads to high thermal stability because the materials have relatively high molecular weights. The substitution of the TPS moiety itself has been reported to result in a high triplet energy level but also lower electron transport ability[17–19]. However, in the case of the DOBNA structure, the electron-transport

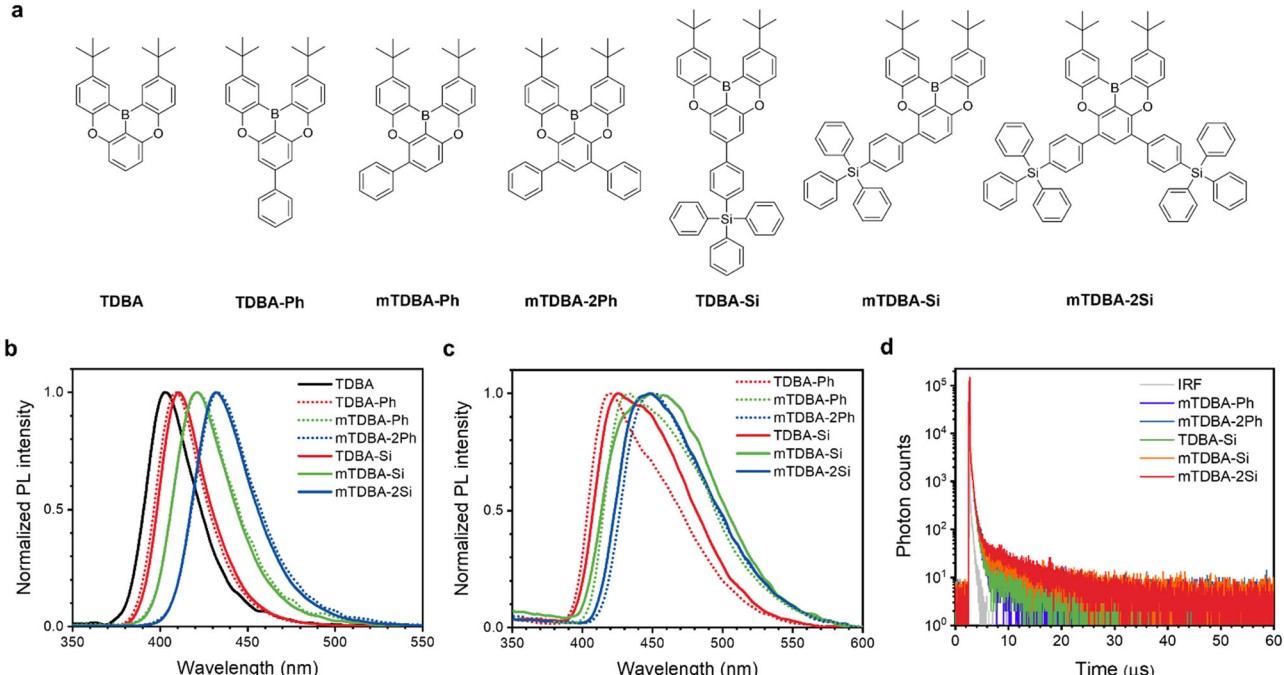

**Fig. 1 | Chemical structure and photophysical property of TDBA-based host materials. a** Chemical structure of the synthesized TDBA-based host materials. **b** Photoluminescence spectra of the TDBA-based host materials in toluene

(0.01 mM). **c** Photoluminescence spectra of the TDBA-based host materials as neat films. **d** Transient photoluminescence decay spectra of the neat films (IRF: instruments response function).

**Table 1 | Summary of the photophysical properties of the TDBA-based materials**

| | Solution | | | Neat Film | | | | | | | | |
|---|---|---|---|---|---|---|---|---|---|---|---|---|
| | $\lambda_{ab}$[a] (nm) | $\lambda_{em}$[a] (nm) | FWHM (nm) | $\lambda_{ab}$[a] (nm) | $\lambda_{em}$[a] (nm) | FWHM (nm) | PLQY (%) | $E_S$ / $E_T$ / $\Delta E_{ST}$[b] (eV) | $\tau_d$ (µs)[c] | HOMO[d] (eV) | LUMO[d] (eV) | $E_g$[d] (eV) |
| TDBA | 383 | 403 | 27 | — | — | — | — | —/ —/ — | — | 6.02 | 2.87 | 3.15 |
| TDBA-Ph | 388 | 408 | 27 | 392 | 418 | 66 | 62 | 3.06/ 2.82/ 0.24 | – | 5.75 | 2.65 | 3.10 |
| mTDBA-Ph | 392 | 421 | 34 | 398 | 434 | 75 | 65 | 2.99/ 2.78/ 0.21 | 1.25 | 5.68 | 2.64 | 3.04 |
| mTDBA-2Ph | 400 | 433 | 37 | 406 | 450 | 70 | 63 | 2.91/ 2.71/ 0.20 | 2.33 | 5.64 | 2.67 | 2.97 |
| TDBA-Si | 389 | 411 | 27 | 393 | 425 | 73 | 66 | 3.02/ 2.77/ 0.25 | 3.53 | 5.73 | 2.64 | 3.09 |
| mTDBA-Si | 393 | 421 | 33 | 398 | 449 | 79 | 61 | 3.01/ 2.73/ 0.28 | 6.65 | 5.71 | 2.66 | 3.05 |
| mTDBA-2Si | 401 | 432 | 35 | 405 | 447 | 75 | 75 | 2.96/ 2.72/ 0.24 | 6.49 | 5.65 | 2.69 | 2.96 |

[a]Maximum wavelength in UV–Vis absorption and photoluminescence spectra.
[b]Singlet and triplet energies measured in the neat film state as an onset value ($\Delta E_{ST}$ = S1–T1).
[c]Delayed lifetime calculated by PL decay for a vacuum-deposited neat film.
[d]HOMO value measured by UV photoelectron yield spectroscopy (AC-2); the LUMO value was calculated from the optical bandgap.

ability can be improved because the boron atom containing an empty $p$ orbital induces an electron-withdrawing effect in the $p_z$–π junction[10]. In addition, host materials that contain a TPS moiety are expected to exhibit a high horizontal molecular orientation effect, leading to an enhanced out-coupling effect because of an increase in molecular length in the long-axis direction[20–23].

The synthetic route to the synthesized host materials is shown in Supplementary Fig. S27. DOBNA was synthesized via a two-step process, and the final materials were synthesized via a Pd-catalyzed Suzuki coupling reaction. All of the synthesized compounds were purified by recrystallization and column chromatography. The synthesized compounds were characterized using nuclear magnetic resonance (NMR) spectroscopy (Supplementary Fig. S28-47), mass spectrometry, and single crystal x-ray diffraction analysis.

**Theoretical calculations**
Density functional theory (DFT) and time-dependent (TD)-DFT calculations were performed to investigate the combination characteristics of the TDBA moiety and various side groups. The dihedral angles of TDBA-Ph, mTDBA-Ph, mTDBA-2Ph, TDBA-Si, mTDBA-Si, and mTDBA-2Si showed similar values at the same position (Supplementary Fig. S2). The electron densities of the HOMO and LUMO of the materials in which phenyl and TPS were substituted at the *meta*- or *para*- position in the TDBA moiety, respectively, were concentrated mainly on the DOBNA moiety. In the case of the materials substituted at the *para*-position, the electron distribution of the phenyl group was mostly localized in the LUMO. By contrast, in the materials substituted at the *meta*- position, the electron distribution of the phenyl group was mostly localized in the HOMO (Supplementary Fig. S3–S9) because the electron density of the *meta*- carbon is higher than that of the *para*-carbon in TDBA[8].

The calculated $\Delta E_{ST}$ values of the synthesized host materials were 0.45–0.52 eV (Supplementary Table S1). In addition, by checking the energy level of the $T_2$ state of these host materials, we confirmed that the energy difference between the $T_2$ level and the $S_1$ level is smaller than the difference between the $T_1$ level and the $S_1$ level (Supplementary Table S2). However, Monkman group reported a well-defined photodynamic mechanism for the RISC process at the $T_1$ and $T_2$ levels[24,25]. On the basis of these findings, the direct RISC process from the $T_2$ level to the $S_1$ level will not occur in the system examined in the present study, and the transition from the $T_1$ level to the $S_1$ level will be the main process.

**Photophysical properties of TDBA-based host materials**
To investigate the photophysical properties, we recorded UV–Visible (UV–Vis) absorption and photoluminescence spectra of the TDBA-based host materials in toluene solution and as vacuum-deposited films (Supplementary Fig. S19, Figs. 1b and 1c). Table 1 summarizes the photophysical properties for the TDBA-based host materials. In the solution state, the TDBA-based host materials exhibited an absorption maximum peak at approximately 383–401 nm and emission was confirmed at 403–433 nm. The UV–Vis spectra of neat films of the TDBA-based host materials showed absorption peaks red-shifted by approximately 4–6 nm relative to the absorption peaks of the corresponding host materials in the solution state (Table 1 and Supplementary Fig. S19). The maximum photoluminescence ($PL_{max}$) intensities of TDBA-Ph, mTDBA-Ph, mTDBA-2Ph, TDBA-Si, mTDBA-Si, and mTDBA-2Si in the film state were observed at wavelengths of 418, 434, 450, 425, 449, and 447 nm, respectively. The FWHM of the absorption peaks of the film-state samples is approximately 33–46 nm broader than that of the absorption peaks of the solution-state samples because the TDBA core has a planar structure. In addition, the strength of the intermolecular interactions can increase because of the mixed conjugation of the Si atom and the phenyl group, as shown in Supplementary Fig. S10-16[26]. Such a mixed conjugation is advantageous in that energy transfer can occur smoothly because an appropriate interaction can be maintained. The PLQYs of TDBA-Ph, mTDBA-Ph, mTDBA-2Ph, TDBA-Si, mTDBA-Si, and mTDBA-2Si in the film state were 62, 65, 63, 65, 61, and 75%, respectively. When a phenyl or TPS group was substituted at the same position in TDBA, similar $S_1$ and $T_1$ levels were observed and the $\Delta E_{ST}$ was approximately 0.20–0.28 eV for all the materials (Table 1). Although $\Delta E_{ST}$ is smaller for $S_1$-$T_2$ than for $S_1$-$T_1$ (Supplementary Table S2), the RISC process occurs only in the $T_1$ state because of rapid internal conversion from $T_2$ to $T_1$. Delayed fluorescence was observed through measurement of the transient photoluminescence of the TDBA-based host materials in the film state (Fig. 1d). The delayed lifetimes of mTDBA-Ph, mTDBA-2Ph, TDBA-Si, mTDBA-Si, and mTDBA-2Si were 1.25, 2.33, 3.53, 6.65, and 6.49 µs, respectively (Table 1 and Supplementary Table S4). In addition, the quantum yield values of the TADF portion of the non-doped film state were in the range of 0.018 to 0.061, which is relatively low compared to the high-performance TADF materials.

**Photophysical properties of v-DABNA-doped films prepared using TDBA-based host materials**
The photophysical properties of TDBA-based host materials doped with 2% v-DABNA, a blue-emitting TADF dopant, were characterized. The v-DABNA dopant exhibits the highest device efficiency among blue-emitting dopants reported to date[6]. Figure 2a and Table 2 show the photoluminescence spectra and the related data for the doped films. The photoluminescence (PL) spectra of all the films show a peak maximum at 469 nm, and the half-width at half-maximum is 18 nm. The PL spectrum of the doped mTDBA-2Si film showed only the residual host emission shoulder peak, indicating imperfect energy transfer from host to dopant. A transient photoluminescence decay curve was recorded to

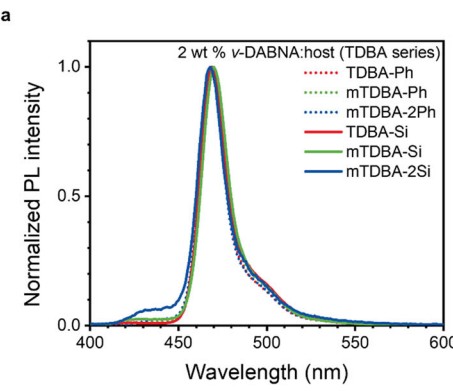
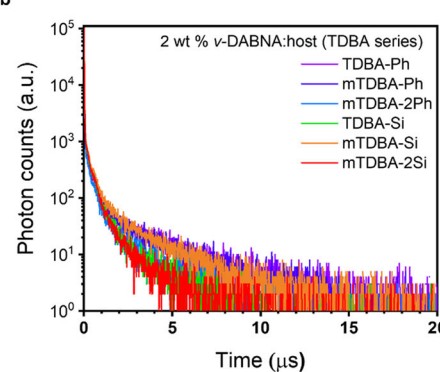

**Fig. 2 | Photophysical properties and schematic of the 2 wt% *v*-DABNA-doped TDBA-based films. a** Photoluminescence spectra. **b** Transient decay spectra of the doped films.

**Table 2 | Photophysical, thermal, and surface properties of doped films**

| | 2 wt% *v*-DABNA-doped film | | | | | $T_g{}^c$ (°C) | $T_d{}^d$ (°C) | RMS[e] (nm) |
|---|---|---|---|---|---|---|---|---|
| | $\theta_{//}{}^a$ (%) | PLQY | $\tau_d{}^b$ (µs) | $\lambda_{max}$/FWHM (nm) | $k_{RISC}$ ($10^5$ s$^{-1}$) | | | |
| TDBA-Ph | 84 | 0.82 | 2.99 | 469/18 | 2.25 | 62 | 255 | 1.65 |
| mTDBA-Ph | 78 | 0.93 | 2.76 | 469/18 | 3.13 | 58 | 230 | 1.42 |
| mTDBA-2Ph | 86 | 0.86 | 1.86 | 468/18 | 3.98 | 93 | 262 | 1.78 |
| TDBA-Si | 91 | 0.82 | 0.96 | 469/18 | 7.01 | 130 | 324 | 0.37 |
| mTDBA-Si | 88 | 0.94 | 1.93 | 469/18 | 4.58 | 122 | 348 | 0.47 |
| mTDBA-2Si | 89 | 0.92 | 0.98 | 468/18 | 8.64 | 180 | 403 | 0.66 |

[a]Horizontal transient dipole orientation ratio.
[b]Delayed lifetime calculated by photoluminescence decay.
[c]Glass transition temperature.
[d]Decomposition temperature.
[e]Root mean square roughness measured by atomic force microscopy (AFM).

confirm the behavior in the excited state between the TDBA-based host materials and the *v*-DABNA dopant (Fig. 2b). To exclude any kinetic trace effect from impurities, excitation spectra were measured in the doped film state (Supplementary Fig. S20). The excitation spectra data of the doped films were matched with the corresponding absorption spectra of the non-doped host materials and doped films. This comparison revealed overlap between the PL spectra of the TDBA-based host materials in the neat film state and the UV absorption spectra of the dopant in the solution state, indicating that energy transfer can readily occur (Supplementary Fig. S21). When TDBA-Ph, mTDBA-Ph, mTDBA-2Ph, TDBA-Si, mTDBA-Si, and mTDBA-2Si were used as hosts in doped films, the average decay times were 2.99, 2.76, 1.86, 0.96, 1.93, and 0.98 µs, respectively (Table 2). The host materials substituted with the TPS moiety showed relatively fast decay, which is advantageous for reducing energy loss via non-radiative channels. This fast decay process, which is faster than the rates of decay recently reported by Prof. Hatakeyama and other groups, means that energy transfer occurs quickly (Supplementary Table S10). Also, based on the results of the transient PL measurements in the doped film state, the exciton decay behavior can be divided into three exponential decay components (Supplementary Table S5). The second ($\tau_2$) and third ($\tau_3$) fitted decay times can be interpreted as corresponding to the dopant and host components, respectively. This interpretation is because not only the FRET rate constants between the synthesized host materials and *v*-DABNA are in the range of $1.07 \sim 2.08 \times 10^8$/s, as shown in Supplementary Table S8, but also the delayed fluorescence of the host is slower than that of the dopant. The anomalous rate constants in Fig. 2b and Supplementary Table S5 can be attributed to exciplex formation of the host-dopant system. The ground state complex was not observed in the doped film, which made host-guest interaction in the excited state highly possible. As reported by Chou group, transient absorption experiments need to

be conducted to examine whether exciplexes form; we will report the excitation dynamics data in a forthcoming paper[27]. Comparing the second exponential decay values ($\tau_2$) of the TPS-based hosts with TPS-free hosts, TDBA-Si and mTDBA-Si showed slightly faster decay than TDBA-Ph and mTDBA-Ph, respectively. When TDBA-Ph, mTDBA-Ph, mTDBA-2Ph, TDBA-Si, mTDBA-Si, and mTDBA-2Si were used as host materials, the PLQY values were 0.82, 0.93, 0.86, 0.82, 0.94, and 0.92, respectively. To understand the interaction between the TDBA-based host materials and *v*-DABNA, we calculated the $k_{RISC}$ values of the doped films (Table 2 and Supplementary Table S6). The $k_{RISC}$ values of the TDBA-Ph, mTDBA-Ph, and mTDBA-2Ph films were 2.25, 3.13, and $3.98 \times 10^5$ s$^{-1}$, respectively. However, the $k_{RISC}$ values of the TDBA-Si, mTDBA-Si, and mTDBA-2Si doped films were 7.01, 4.58, and 8.64 $\times 10^5$ s$^{-1}$, respectively. The compounds with the TPS moiety thus showed RISC speeds 1.5 to 3.1 times faster than those of the compounds with a phenyl group. In a two-component system, the synthesized host materials may facilitate the FRET process to the dopant through RISC activation of the host $T_1$ level.

To confirm that efficient energy transfer occurs between the TPS-based host materials and *v*-DABNA, we measured the energy transfer values of the Stern-Volmer equation (Supplementary Table S7). The $k_q$ values of the three TPS-based host materials are larger than those of the corresponding TPS-free host materials. Also, calculation of the $k_{FRET}$ values confirmed that two materials containing the TPS moiety (mTDBA-Si and mTDBA-2Si) show faster $k_{FRET}$ than the TPS-free materials (Supplementary Table S8)[28].

The transient dipole orientation ratio was measured through variable-angle spectroscopic ellipsometry and angle-dependent *p*-polarized photoluminescence measurements of the doped films (Table 2 and Supplementary Fig. S22). Because the concentration of dopant in the host–dopant system was very low (e.g., 2%), the

molecular structure of the host is important from the viewpoint of molecular orientation[22–25]. When TDBA-Ph, mTDBA-Ph, and mTDBA-2Ph with phenyl groups were used as hosts, the horizontal transient dipole orientation ratios were 84, 78, and 86%, respectively. However, when TDBA-Si, mTDBA-Si, and mTDBA-2Si with the TPS moiety were used as hosts, the values increased to 91, 88, and 89%, respectively. That is, the introduction of the TPS moiety increased the out-coupling efficiency by imparting a horizontal orientation to the transient dipole of the dopant material.

## Surface morphology and thermal properties

To confirm the surface morphology of the TDBA-based host materials, we characterized them using optical microscopy (OM) and atomic force microscopy (AFM). The materials with the TPS moiety showed relatively good film uniformity but in the case of the TDBA, TDBA-Ph, and mTDBA-Ph films, crystallization occurred simultaneously (Supplementary Fig. S23-24). Such rapid crystallization can lead to a decrease in device efficiency because it induces adverse effects such as a rough surface, charge trapping at the interface, and an increase in the drift current. To confirm the glass-transition temperature ($T_g$), crystallization temperature ($T_c$), melting temperature ($T_m$), and decomposition temperature ($T_d$) of the TDBA-based host materials, we carried out thermogravimetric analysis (TGA) and differential scanning calorimetry (DSC) (Supplementary Fig. S25-26). The $T_d$ values corresponding to 5 wt% loss of TDBA, TDBA-Ph, mTDBA-Ph, mTDBA-2Ph, TDBA-Si, mTDBA-Ph, and mTDBA-2Ph were 184, 255, 230, 262, 324, 348, and 403 °C, respectively. The $T_g/T_c/T_m$ of the TDBA-Ph, mTDBA-Ph, mTDBA-2Ph, TDBA-Si, mTDBA-Ph, and mTDBA-2Ph host materials were 62/107/217 °C, 58/126/175 °C, 93/139/224 °C, 130/213/277 °C, 122/154/220 °C, and 180/262/377 °C, respectively (Table 2). The materials with the phenyl moiety showed relatively low $T_g$ and $T_c$ values, which means that crystallization occurs relatively easily. On the other hand, the materials with the TPS moiety exhibit high $T_g$ and $T_c$ values, which positively affects device operation as well as device fabrication.

## OLED device characterization and performance

Doped OLED devices were fabricated using the synthesized TDBA-based host materials in the emitting layer and the EL performance of the devices was characterized. The device configuration was ITO/NPB (40 nm)/TCTA (15 nm)/mCP (15 nm)/host material: 2% *v*-DABNA (20 nm)/TmPyPB (40 nm)/LiF(1 nm)/Al (200 nm) (see Experimental section-OLED device).

Figure 3a shows a proposed energy-transfer mechanism between the host and *v*-DABNA dopant used in this study.

The energy-level diagrams of the materials are shown in Fig. 3b. All devices fabricated using the host materials exhibited the same EL maximum value of 465 nm, FWHM of 18 nm, and Commission Internationale de l'Eclairage (CIE) coordinates of (0.13, 0.10) (Fig. 3f and Table 3). A comparison of the EL performance of each device with different host materials reveals that the devices containing the TPS moiety exhibit a relatively high maximum brightness as well as a slightly low turn-on voltage. The devices exhibit different EL performances depending on the inclusion or exclusion of the TPS moiety (Table 3 and Fig. 3c, d). The current efficiencies (CEs) of the devices fabricated using TDBA-Si, mTDBA-Si, or mTDBA-2Si as the host were 38.1/36.2/31.8 cd A$^{-1}$, 27.9/27.5/24.1 cd A$^{-1}$, and 50.3/37.5/21.5 cd A$^{-1}$, respectively, at maximum luminance/500 cd m$^{-2}$/1000 cd m$^{-2}$. The EQE values were 36.2/35.0/31.3%, 27.3/26.9/24.1%, and 38.1/28.8/16.6%, respectively. At 1,000 cd m$^{-2}$, these devices showed a 2.8–3.8 times greater CE and a 2.3–3.0 times greater EQE compared with the devices fabricated using TDBA-Ph, mTDBA-Ph, and mTDBA-2Ph. In addition, the device prepared using TDBA-Si as the host material exhibited an EQE maximum value of 36.2% and 31.3% efficiency even at a high brightness of 1,000 cd m$^{-2}$, resulting in a smaller roll-off (4.9%) compared with that of devices prepared with conventional MR-TADF

materials. In particular, among all reported TADF materials, those prepared in the present work are the first to exhibit not only blue emission with a CIE *y* value smaller than 0.15, which is commonly used commercially, but also an EQE greater than 30% under the high-luminance condition of 1000 cd m$^{-2}$ (Fig. 4 and Supplementary Table S10). The high efficiency and low roll-off might be due to the prevention of non-radiative processes. The more severe roll-off observed for mTDBA-2Si compared to TDBA-Si and mTDBA-Si may be due to the higher imbalance in charge mobility of holes and electrons of mTDBA-2Si (Supplementary Table S9). Specifically, the charge balance value, which is the hole mobility of the hole-only device (HOD) divided by the electron mobility of the electron-only device (EOD) in the doped devices, of TDBA-Si, mTDBA-Si, and mTDBA-2Si was 1.05, 1.29, and 4.00, respectively. The imbalanced carrier mobility of holes and electrons in mTDBA-2Si can cause severe EQE roll-off in device performance.

Possible explanations for the enhanced EL performance of the TPS-based host materials compared with that of the TPS-free host materials are as follows. First, the enhanced EL performance can be explained by the energy transfer from the TPS-based host materials to the dopant. As mentioned above in the discussion of PL, larger energy transfer values in the doped TPS-based films were confirmed by the results of Stern-Volmer experiments, and these larger energy transfer values could cause not only decreased triplet-exciton loss but also increased EL performance. Second, the enhanced EL performance can be explained by the more balanced carrier mobility of the TPS-based host materials compared to TPS-free host materials (Supplementary Table S9). Third, the enhanced EL performance can be explained by the high horizontal molecular dipole orientation. As previously mentioned, the horizontal orientation ratios of TDBA-Ph, mTDBA-Ph, and mTDBA-2Ph were 84%, 78%, and 86%, while TDBA-Si, mTDBA-Si, and mTDBA-2Si were 91%, 88%, and 89%, indicating an increment of 7%, 10%, and 3%, respectively. As reported by Prof. Kwon's group, the high horizontal orientation ratio is the basis for TDBA-Si exhibiting an EQE of 36% in simulations[10]. Fourth, the enhanced EL performance of the TPS-containing host materials may be matched with their excellent thermal stability and surface properties (Supplementary Fig. S23–26). TDBA-Si, mTDBA-Si, and mTDBA-2Si exhibit high $T_g$ values in the range of 122–180 °C, whereas mTDBA-2Ph, TDBA-Ph and mTDBA-Ph exhibit $T_g$ values of 93, 62 and 58 °C, respectively. Thus, the $T_g$ increment on going from TDBA-Ph and mTDBA-Ph to TDBA-Si and mTDBA-Si is 68 and 64 °C, respectively. As a result, the TPS-based host materials maintain an amorphous state well because they do not easily undergo molecular changes. The stable surface morphology of the deposited films maintains the device efficiency. Additionally, the long intermolecular distances in host materials containing the bulky TPS group may prevent triplet-triplet annihilation (TTA) and/or singlet-triplet annihilation (STA), leading to increased EQE[29].

## Discussion

Three DOBNA derivatives with a TPS moiety—TDBA-Si, mTDBA-Si, and mTDBA-2Si—were designed and synthesized as MR-TADF-type host materials. These synthesized host materials have a high $T_1$ level of 2.71–2.82 eV; they are therefore suitable hosts for the blue-emitting MR-TADF dopant *v*-DABNA, and a high PLQY and high horizontal orientation were induced in the doped films. In addition, upon introduction of the TPS moiety, effective energy transfer to the dopant was observed. When TDBA-Si was used in an OLED device, the device exhibited a remarkably small roll-off, with an EQE of 31.3% at 1000 cd m$^{-2}$ as well as an ultra-high efficiency of 36.2% EQE$_{max}$ at CIE coordinates (0.13, 0.10). This performance is likely attributable to the fast and efficient FRET to the dopant resulting from the substitution effect of the TPS in the host and to the prevention of the non-radiative process of the triplet exciton. The molecular design concept in which TPS is introduced to obtain an MR-TADF-type derivative is expected to facilitate the development of TADF hosts for high-performance blue OLEDs.

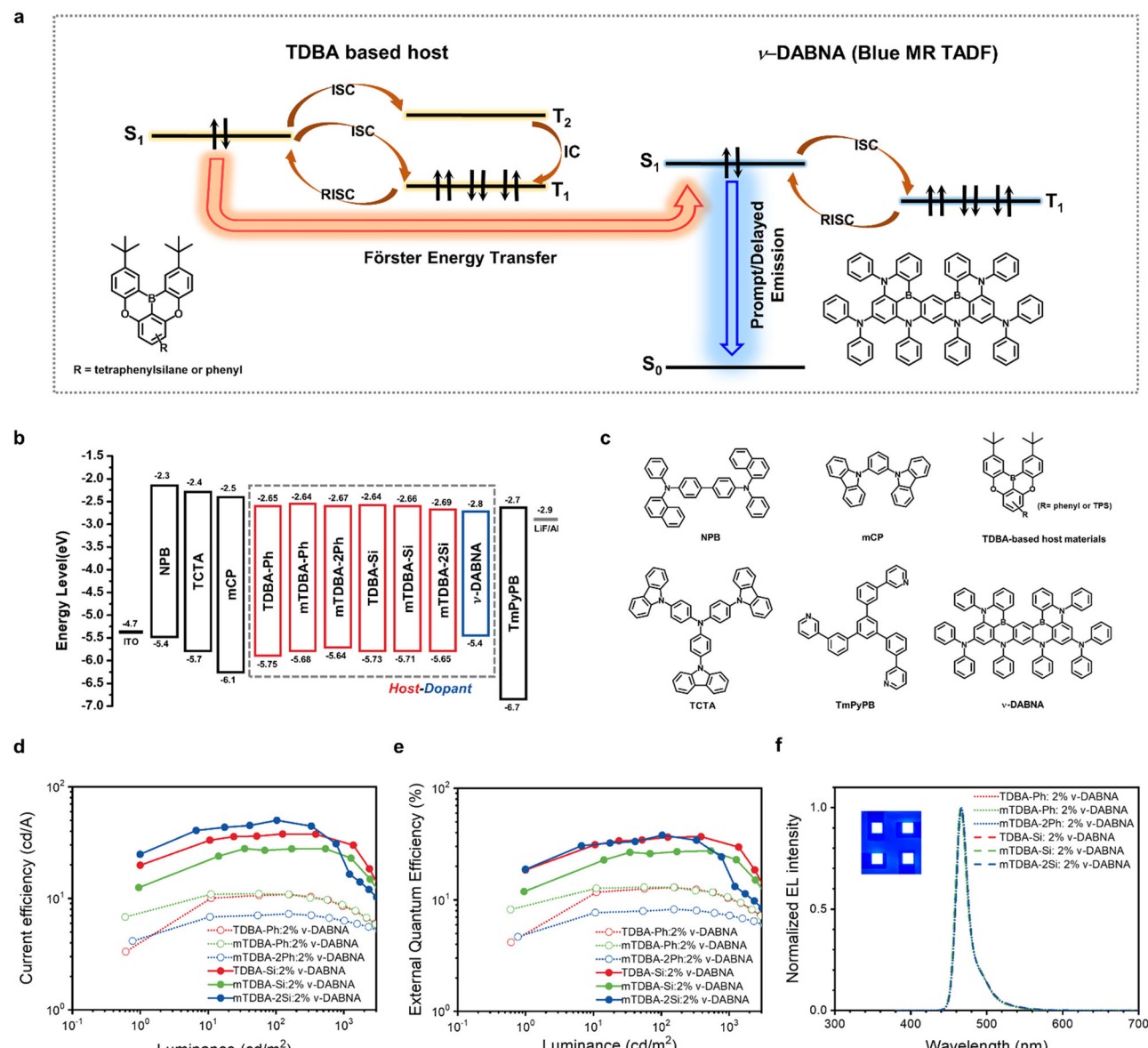

**Fig. 3 | EL performance of TDBA-based host materials containing Si atom.**
**a** Schematic of the conceived energy-transfer mechanism for the host and the *v*-DABNA dopant. **b** Energy-level diagram of doped OLED devices. **c** Molecular structures used in each layer. **d** Current efficiency (CE)−luminance (L) curves. **e** External quantum efficiency (EQE)−L curves. **f** EL spectra (inset: OLED driving image at 7 V).

## Methods

### Materials and General information

All commercially available chemicals were purchased from Sigma Aldrich, Tokyo Chemical Industry (TCI), and Alfa Aesar. Purchased chemicals were used as received without further purification. All experimental glassware and equipment were dried in a convection oven before use. All reactions described in here were conducted in an argon saturated atmosphere and monitored by thin layer chromatography (TLC) technique using TLC plate (silica gel 254, Merk Co.). Detailed purification conditions are described in the following synthetic detail parts. All synthesized compounds were characterized by [1]H-NMR, [13]C-NMR, and single crystal x-ray analysis. [1]H-NMR and [13]C-NMR spectra were acquired using Bruker, Avance-300 and 500 NMR spectrometer.

The highest occupied molecular orbital (HOMO) absolute energy levels were determined by photoelectron yield spectroscopy (Riken Keiki AC-2). (Supplementary Fig. S18). The lowest unoccupied molecular orbital (LUMO) energy levels were evaluated from the HOMO energy levels and optical bandgap, i.e., LUMO (eV) = HOMO (eV) +

bandgap (eV). Thermogravimetric analysis (TGA) was performed using Discovery TGA (TA instruments) at a rate of 10 °C min$^{-1}$ in a N$_2$ atmosphere. Differential scanning calorimetry (DSC) measurements were performed under a N$_2$ atmosphere using a Q-1000 (TA instruments).

### X-ray Crystallography

To obtain a better understanding of the molecular structures in the solid-state samples, we conducted single-crystal X-ray diffraction analyses. Single crystals were prepared by the solvent-diffusion crystal growth method using dichloromethane/ethanol (1:2 v/v) for TDBA-Ph, TDBA-Si, mTDBA-Ph, and mTDBA-Si. The crystal structures and crystallographic data are shown in Supplementary Fig. S17 and Supplementary Data 1–8. The results confirm that the dihedral angles between the TDBA and side groups were similar to the calculated values. Single crystal structures were analyzed by the Instrumental Analysis Center, Gyeongsang National University using APEX II Ultra. Crystal data for TDBA-Ph showed a structure of the space group P21/c of the monoclinic crystal system with unit cell parameters a = 16.86 Å,

**Table 3 | EL performance of 2 wt% *v*-DABNA-doped OLED devices according to the host materials**

| EMLs | $T_{on}$[a] (V) | $L_{max}$ (cd m$^{-2}$) | CE[b] (cd A$^{-1}$) | | | EQE[c] (%) | | | CIE[d] (x, y) | EL$_{max}$ /FWHM (nm) |
|---|---|---|---|---|---|---|---|---|---|---|
| | | | Max. | 500 cd m$^{-2}$ | 1000 cd m$^{-2}$ | Max. | 500 cd m$^{-2}$ | 1000 cd m$^{-2}$ | | |
| TDBA-Ph: *v*-DABNA | 3.10 | 4400 | 10.8 | 9.80 | 8.48 | 12.9 | 11.8 | 10.3 | (0.13,0.10) | 465/18 |
| mTDBA-Ph: *v*-DABNA | 3.09 | 4500 | 10.9 | 9.81 | 8.48 | 13.0 | 11.8 | 10.4 | (0.13,0.10) | 465/18 |
| mTDBA-2Ph: *v*-DABNA | 3.06 | 5900 | 7.24 | 6.82 | 6.34 | 8.21 | 7.75 | 7.24 | (0.13,0.10) | 465/18 |
| TDBA-Si: *v*-DABNA | 3.00 | 7300 | 38.1 | 36.2 | 31.8 | 36.2 | 35.0 | 31.3 | (0.13,0.10) | 465/18 |
| mTDBA-Si: *v*-DABNA | 3.02 | 7100 | 27.9 | 27.5 | 24.1 | 27.3 | 26.9 | 24.1 | (0.13,0.10) | 465/18 |
| mTDBA-2Si: *v*-DABNA | 3.00 | 6300 | 50.3 | 37.5 | 21.5 | 38.1 | 28.8 | 16.6 | (0.13,0.10) | 465/18 |

[a]Turn-on voltage at 1 cd m$^{-2}$.
[b]Current efficiency.
[c]External quantum efficiency.
[d]Commission Internationale de l'Eclairage.

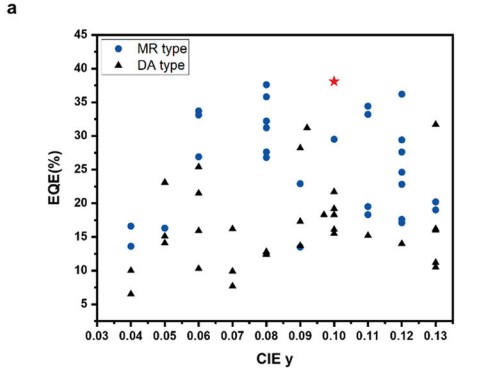
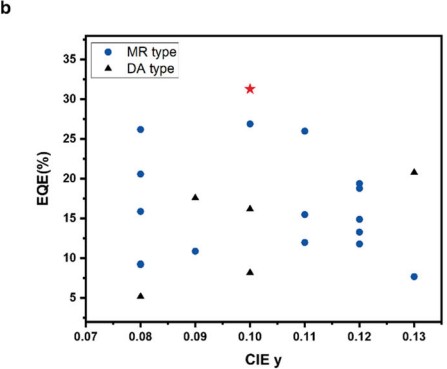

**Fig. 4 | Summary of representative deep blue (CIE y < 0.15) OLEDs. a** EQE$_{max}$ (TDBA-2Si). **b** EQE at 1,000nit (TDBA-Si).

b = 13.48 Å, c = 22.89 Å, α = γ = 90 °, and β = 103.05(3) °. mTDBA-Ph also showed the structure of the space group P21/c of the monoclinic crystal system, with unit cell parameters a = 15.73 Å, b = 6.16 Å, c = 25.69 Å, α = γ = 90 °, and β = 95.53 (10) °. TDBA-Si exhibited a structure of the space group P21/c of the monoclinic crystal system with unit cell parameters a = 7.41 Å, b = 17.78 Å, c = 15.18 Å, α = γ = 90 °, and β = 98.61(2) °. mTDBA-Si had the structure of the space group P-1 of the triclinic crystal system, with unit cell parameters a = 9.69 Å, b = 13.96 Å, c = 14.93 Å, α = 86.88(10)°, β = 83.41(10) °, and γ = 86.48 °. CCDC (2215050), (2215051), (2215052), (2215053) for TDBA-Ph, mTDBA-Ph, TDBA-Si, and mTDBA-Si, respectively.

## Photophysical properties

Solution samples were prepared with toluene at a concentration of 0.01 mM. The neat film was deposited on quartz at a rate of 1 angstrom per second under vacuum. The UV-Vis absorption spectrum was obtained using a Lambda 1050 UV/Vis/NIR spectrometer (PerkinElmer). Photoluminescence (PL) spectra were recorded using the Photon Technology International QM-40. Absolute photoluminescence quantum yield (PLQY) and transient quantaurus-QY (Hamamatsu) were obtained. The low-temperature photoluminescence spectrum was measured at 77 K using the Jasco FP-6500. Photoluminescent decay traces were obtained through time-correlated single-photon coefficient (TCSPC) technology using PicoQuant, FluoTime 250 instruments (PicoQuant, Germany). A 377 nm pulse laser was used as an excitation source and data analysis was performed using the exponential fitting model of the FluoFit software.

## Determination of exciton dynamic parameters related to TADF

Exciton dynamic parameters for v-DABNA in TDBA-based host materials (2 wt% doped film) at 298 K. All parameters are calculated according to Eqs. (1)–(7).

$$k_F \quad k_F = \Phi_F / \tau_F \tag{1}$$

$$k_{IC} \quad \Phi = k_F / (k_F + k_{IC}) \tag{2}$$

$$k_{ISC} \quad \Phi_F = k_F / (k_F + k_{IC} + k_{ISC}) \tag{3}$$

$$\Phi_{IC} \quad \Phi_{IC} = k_{IC} / (k_F + k_{IC} + k_{ISC}) \tag{4}$$

$$\Phi_{ISC} \quad \Phi_{ISC} = 1 - \Phi_F - \Phi_{IC} = k_{ISC} / (k_F + k_{IC} + k_{ISC}) \tag{5}$$

$$k_{TADF} \quad k_{TADF} = \Phi_{TADF} / \Phi_{ISC} \tau_{TADF} \tag{6}$$

$$k_{RISC} \quad k_{RISC} = k_F k_{TADF} \Phi_{TADF} / k_{ISC} \Phi_F \tag{7}$$

## DFT/TD-DFT calculation

Quantum chemical calculations based on density functional theory (DFT) were performed using a Gaussian 09 program. Theoretical

calculations with geometric optimization, single point energy, HOMO and LUMO distributions, lowest singlet excited state ($S_1$), and lowest triplet excited state ($T_1$) were performed at density function theory (DFT) and time-dependent DFT (TD-DFT) levels using B3LYP functioning with 6-31 G (d,p) basis sets. Spin-orbit coupling matrix elements were performed using the ORCA program with the same basis sets (Supplementary Data 9–22).

The calculated HOMO energy levels of TDBA, TDBA-Ph, mTDBA-Ph, mTDBA-2Ph, TDBA-Si, mTDBA-Si, and mTDBA-2Si were −5.49, −5.49, −5.37, −5.28, −5.50, −5.38, and −5.31 eV, respectively; the calculated LUMO energy levels were −1.59, −1.67, −1.61, −1.63, −1.70, −1.63, and −1.98 eV, respectively. The order of the calculated bandgaps ($\triangle E_{H-L}$) was TDBA (3.90 eV) > TDBA-Ph (3.83 eV) > TDBA-Si (3.80 eV) > mTDBA-Ph (3.75 eV) ≈ mTDBA-Si (3.76 eV) > mTDBA-2Ph (3.65 eV) = mTDBA-2Ph (3.65 eV). The bandgaps measured experimentally showed the same tendency as the calculated values (Supplementary Fig. S3–S9, Table S1, and Table 1).

## Device fabrication

Device configuration: ITO/NPB (40 nm)/TCTA (15 nm)/mCP (15 nm)/host materials: 2% v-DABNA (20 nm)/TmPyPB (40 nm)/LiF (1 nm)/Al (200 nm). N,N′-Bis(naphthalen-1-yl)-N,N′-bis(phenyl)-benzidine (NPB), 4,4′,4-Tris(carbazol-9-yl)triphenylamine (TCTA) used as an injection layer. 1,3-Bis(carbazol-9-yl)benzene (mCP) was used as both a hole transporting layer and an electron blocking layer. 1,3,5-Tris(3-pyridyl-3-phenyl)benzene (TmPyPB) was used as both an electron transporting layer and a hole blocking layer. As a dopant for the emitting layer, v-DABNA, a multiple resonance TADF material reported by Hatakeyama et al. was used. As the host material, synthesized TDBA-based host materials were applied to confirm the relationship between host and dopant. For the EL devices, all organic layers were deposited under $10^{-6}$ torr, with a rate of deposition of 1 Å/s to give an emitting area of 4 mm². The LiF and aluminum layers were continuously deposited under the same vacuum conditions. The current-voltage-luminance (I-V-L) characteristics of the fabricated EL devices were obtained with a Keithley 2400 electrometer. Light intensities were obtained with a Minolta CS-1000A. To calibrate the EQE values considering the angular dependence, emission angular distributions were also measured. The operational stabilities of the devices were measured under encapsulation in a glovebox.

## Reporting summary

Further information on research design is available in the Nature Portfolio Reporting Summary linked to this article.

## Data availability

The authors declare that all data supporting the findings of this study are available within the paper and its supplementary information files or from the corresponding author upon request. Source data for Figs. 1, 2, and, 3 are provided in the figshare (https://doi.org/10.6084/m9.figshare.23984025). The X-ray crystallographic data for structures reported in this study have been deposited at the Cambridge Crystallographic Data Center (CCDC). CCDC numbers for TDBA-Ph, mTDBA-Ph, TDBA-Si, and mTDBA-Si are CCDC 2215050-2215053. These data can be obtained free of charge from The Cambridge Crystallographic Data Center via www.ccdc.cam.ac.uk/structures/.

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

## Acknowledgements

The work at Seoul National University was supported by a research grant of Samsung Display Co. [RIAM0417- 20220144]. The research at Kyung Hee University was supported by the Basic Science Research Program through the National Research Foundation of Korea (NRF) funded by the Ministry of Education (2020R1A6A1A03048004). This work was partly supported by the GRRC program of Gyeonggi province (GRRCKYUN-GHEE2023-B01). This research was supported by Basic Science Research Capacity Enhancement Project through Korea Basic Science Institute (National research Facilities and Equipment Center) grant funded by the Ministry of Education (No. 2019R1A6C1010052).

## Author contributions

D.P. and S.K. synthesized and characterized the objective molecules, fabricated the devices, discussed the results, and prepared the manuscript. C.H.R., B.H.J., and S.J. carried out the theoretical calculations. J.L., T.N.L. and M.C.S. discussed the device results, M.E.J. and C.C. provided a funding source, and J.P. and S.Y.P. conceived the project, discussed the results, and prepared the manuscript.

## Competing interests

The authors declare no competing interests.
