## [Peer Review File · Nature Communications]

High-performance blue OLED using multiresonance thermally activated delayed fluorescence host materials containing silicon atomsREVIEWER COMMENTS

Reviewer #1 (Remarks to the Author):

Park and co-workers have reported a total of six TDBA-based host materials, namely, TDBA-Ph/Si, mTDBA-Ph/Si and mTDBA-2Ph/2Si, of which the photophysical properties and device performance in combination with v-DABNA were discussed based on the absence or presence of the silicon moiety. Additionally, the authors performed TD-DFT calculations, thermal stability tests and X-ray crystallography to support the device high performances. This, together with increase in electroluminescent stability, provides a new series of host choices, which deserves its publication on the journal Nature Communication. However, throughout the text, some issues were not well explained by the authors. Therefore, further revisions on this manuscript are required prior to its consideration of publication. Comments and suggestion are listed below:

Major revisions:

1. The authors mentioned the usage of hot triplet state several times throughout the text. However, from a fundamental point of view, it is not quite possible for such RISC to occur from highly excited triplet states unless solid and definitive evidences are provided, which apparently are not the case during the past few years. All relevant researches cite the original paper without providing any direct evidence, which is not a right scientific progress at all. The calculated SOC values in the supporting information were all less than 1 cm^{-1} , for which the maximal rate of ISCs should lie in the nanosecond scale. Secondly, for any highly excited triplet state undergoing RISC, the thermal relaxation due to internal conversion should be dominant. Therefore, the slow RISC rate together the supposed to be ultrafast $T_n \rightarrow T_1$ internal conversion ($\sim \text{ps}$ for large organic molecules) should lead to small or negligible S_1 repopulation by hot triplet states. Before any other groups (except for the original Ma's group) experimentally proving the involvement of RISC from the highly excited triplet states, which is not the case to my knowledge, the authors should town down or remove the related assignments and cited references.
2. I could not find any excitation spectra throughout the text. In order to exclude any kinetic trace from impurities, the authors should provide excitation spectra of all 6 doped films and check whether they resemble to their absorption spectra.
3. Based on the measured delayed population decays ($1 \sim 3 \mu\text{s}$), the authors claimed that the RISC of v-DABNA is activated by the newly designed hosts. However, comparisons on the delayed lifetimes between v-DABNA doped in different hosts were not mentioned by the authors. The delayed lifetimes should also be added in Table S5.

4. Continuing suggestion #3, there is surely significant host-guest interaction (either ground state or excited state) that results in such anomalous short delayed lifetimes. For merely v-DABNA dissolved in dilute solution (or other MR-TADF dopants), TADF is usually not feasible due to the high ΔE_{ST} . However, I cannot see any description on the host-guest interaction or related references. The authors are encouraged to perform steady state or transient absorption on v-DABNA doped films to capture any plausible ground state complex or exciplex formation. To my knowledge, a typical example was recently published in Nature Photonics 15.10 (2021): 780-786.

5. Continuing suggestion #4, the band alignment between host and dopant materials will be the determining factor for the device performance if host-guest interactions exist. Hence, the authors should experimentally access the HOMO and LUMO energy with photoelectron spectroscopy or cyclic voltammetry.

6. The scheme in Figure 2c is incorrect considering from a kinetic point of view. To initiate, when monitoring at the wavelength of dopant's emission, we could observe a tri-exponential decay as seen in Figure 2b. If the authors successfully derive the mathematical relations for this type of system, one will find that the energy transfer rate actually refers to the second lifetime but not the third (longest) lifetime. Additionally, the third lifetime corresponds to the population decay of dopant's triplet state. Therefore, the energy transfer rate for the silicon based hosts are not faster than that of phenyl-based hosts. Figure 2a also showed an emission shoulder for mTDBA-2Si, which indicates its inferior transfer efficiency comparing to other hosts.

7. Continuing suggestion #6, equation (1)~(7) are used to access kinetic parameters when TADF is achieved by the dopant molecule itself. As the authors showed, an energy transfer pathway also involves in the excited state dynamics. As a result, all kinetic parameters deduced in this manuscript have to be re-evaluated. The authors should simulate the kinetic parameters again in consideration of the energy transfer.

8. How did the authors confirm that the FRET takes place from host's hot triplet state to dopant's singlet state? This assignment is fundamentally weird! The authors are recommended to show the spectral overlap between the fluorescence spectrum of host and the absorption spectrum of dopant in the supporting information.

9. In Table S5, the EQE roll off for mTDBA-2Si is much more severe than that of TDBA-Si and mTDBA-Si. Please explain.

Minor revisions:

1. The reported ΔE_{ST} s were calculated by subtracting the S1 energy with T1 energy, which the authors obtained by subtracting the spectral onset of room temperature fluorescence with that of 77 K

phosphorescence. Alternatively, the authors are recommended to calculate the ΔE_{ST} s by subtracting the spectral onset of 77 K fluorescence with that of 77 K phosphorescence.

2. In the Method part, there is a typo in Equation 5. It should be $\Phi_{ISC} = 1 - \Phi_F - \Phi_{IC}$.

3. The authors claimed that aggregations were avoided by inserting the silicon moiety. Accordingly, the packing environment as well as the intermolecular distances should be provided in a figure.

4. Please unify the format of Reference section.

Reviewer #2 (Remarks to the Author):

In this paper, the authors reported three MR-TADF molecules by using the silicon atom. The devices fabricated by using them as host materials showed excellent performance with the EQEs up to 36.2%. But, I think the primary reason of achieving excellent performance is caused by the high device technology level and by using the star MR-TADF emitter, *v*-DABNA, which has been explored widely as the emitter for high-performance OLEDs with the EQE exceeds 30%. Moreover, the molecular design is not novel (Angew. Chem. Int. Ed. 2015, 54, 13581–13585) and the mechanism of the energy transfer proposed in this paper has no definitive proof. Therefore, I do not think that this paper can be published in Nature Communications. Here are some suggestions for authors, hoping to improve the quality of the manuscript and then published on a more specialized journal in the future.

1. The authors should provide more comprehensive sets of experiments and theoretical calculations supporting the fast energy transfer from the host materials containing the silicon atom to dopant.

2. For the mechanism of the energy transfer in fig 2c, here is a certain possibility of the Dexter energy transfer from host to dopant because the RISC rate of the host materials is not fast enough. Moreover, fig 2c shows the mechanism of the energy transfer under the electro-excitation, not for the photo-excitation, so it is not suitable in fig 2, but should be in fig 3 instead.

3. I do not think the silicon is a heavy atom, so here there is no strong evidence of having heavy-atom effect. The values of the SOC matrix elements of the molecules in this paper are too small, which are not comparable with the ones for those molecules containing the sulfur or selenium atom (Angew. Chem. Int. Ed. 2022, e202205684; Nat. Photonics 2022, 16, 803–810).

Reviewer #3 (Remarks to the Author):

The manuscript by Jongwook Park et al. reports TDBA-based host materials possessing one or two tetraphenylsilyl (TPS) groups. The devices using the host materials and v-DABNA exhibited high EQE (up to 36.2%) and small efficiency roll-off (31.3% at 1000 cd/m²). This is one of the best performances of blue MR-TADF-OLEDs, as summarized in Figure 4. However, the claims and discussion regarding the heavy atom effect of the TPS group are not well supported by the experimental and computational results. It should be re-evaluated after significant and careful revisions. Detailed comments on the authors' claims are listed below.

Authors' claim: Possible explanations for the enhanced EL performance of the TPS-containing host materials compared with that of the phenyl-substituted host materials are as follows. First, the enhanced EL performance can be explained by the rapid energy transfer of the three TPS-substituted materials to the dopant. In the v-DABNA-doped films of the three materials with the TPS moiety, the delayed lifetime was as much as three times shorter than that of the doped films of the phenyl-substituted materials, which means that the triplet energy level can be easily utilized. This availability of the triplet level is expected to minimize the triple-exciton loss of the host and increase the energy transfer to v-DABNA. As a result, the external heavy-atom effect of Si in the host material can increase the SOC and the kRISC of the v-DABNA dopant, which increases the EQE value of the doped device. The small roll-off characteristics of the devices under high luminance and high electric fields are also attributed to this fast energy transfer.

Comment: I do not understand this claim. Why does the host-to-dopant energy transfer process correlate with the delayed lifetime? The shorter delayed lifetime is simply attributed to higher kRISC of the dopant in the PL process, which is independent of the energy transfer in the EL process. The authors should discuss the PL and EL processes separately to clarify the point. Moreover, in Figure 3c, the efficiency roll-offs for three TPS-based host materials appear comparable to those for three TPS-free host materials. In particular, mTBDA-2Si, which exhibits the shortest delayed lifetime and highest kRISC value, exhibited the largest roll-off, which is inconsistent with the claim. The authors should discuss on the difference of EQE at lower current density rather than the roll-off at higher current density. The difference at lower current density may be simply explained by exciton formation efficiency. I assume that the TPS-free host materials have lower carrier mobilities than the three TPS-based host materials because of their smaller π -skeleton with the bulky tert-butyl substituents. Therefore, it is strongly recommended to determine the carrier mobilities of the neat films of the host materials with/without the dopant.

Authors' claim: Second, the enhanced EL performance can be explained by the role of the hot triplet excited state. The existence of a T_n state similar to the S₁ energy level might improve the EL performance through the activated RISC process.

Comment: This is possible, but not supported by the experimental data. As shown in Figure 1 and Table 1, the neat films of the TPS-free host materials showed shorter delayed lifetimes than those of the three TPS-based host materials. This indicates that the RISC process at the former is faster than those at the latter in the OLEDs. Moreover, the SOC and TD-DFT calculations in Figure S2–S8 do not clearly support this claim. Clear and careful arguments are required to convince the readers.

Authors' claim: Third, the enhanced EL performance of the TPS-containing host materials can be explained by their excellent thermal stability and surface properties (Supplementary Fig. S19-22). TDBA-Si, mTDBA-Si, and mTDBA-2Si exhibit a high T_g in the range 122–180 °C; they therefore do not easily undergo molecular changes because they maintain an amorphous state. The surface morphology of the deposited films will affect the device efficiency.

Comment: I do not agree with this claim. A T_g of 93 °C (mTBDA-2Ph) should be sufficient for the device fabrication and IVL measurement. Crystallization of the host material during the process significantly changes the EL properties at certain data points.

Authors' claim: Fourth, the enhanced EL performance can be explained by the high horizontal molecular dipole orientation. As previously mentioned, the horizontal orientation ratio of the three materials with the TPS moiety is very high (approximately 88–91%). As reported by Kwon's group, the highly horizontal orientation ratio is the basis for TDBA-Si exhibiting an EQE of 36% in simulations.

Comment: As shown in Figure 18, the horizontal orientation ratio for the TPS-based host materials (88-91%) is slightly higher than those (78-86%) of the TPS-free host materials. However, these small differences do not account for the experimental results.

Reviewer #1 (Remarks to the Author):

Park and co-workers have reported a total of six TDBA-based host materials, namely, TDBA-Ph/Si, mTDBA-Ph/Si and mTDBA-2Ph/2Si, of which the photophysical properties and device performance in combination with ν -DABNA were discussed based on the absence or presence of the silicon moiety. Additionally, the authors performed TD-DFT calculations, thermal stability tests and X-ray crystallography to support the device high performances. This, together with increase in electroluminescent stability, provides a new series of host choices, which deserves its publication on the journal Nature Communication. However, throughout the text, some issues were not well explained by the authors. Therefore, further revisions on this manuscript are required prior to its consideration of publication. Comments and suggestion are listed below:

Major revisions:

1. The authors mentioned the usage of hot triplet state several times throughout the text. However, from a fundamental point of view, it is not quite possible for such RISC to occur from highly excited triplet states unless solid and definitive evidences are provided, which apparently are not the case during the past few years. All relevant researches cite the original paper without providing any direct evidence, which is not a right scientific progress at all. The calculated SOC values in the supporting information were all less than 1 cm⁻¹, for which the maximal rate of ISCs should lie in the nanosecond scale. Secondly, for any highly excited triplet state undergoing RISC, the thermal relaxation due to internal conversion should be dominant. Therefore, the slow RISC rate together the supposed to be ultrafast T_n→T₁ internal conversion (~ ps for large organic molecules) should lead to small or negligible S₁ repopulation by hot triplet states. Before any other groups (except for the original Ma's group) experimentally proving the involvement of RISC from the highly excited triplet states, which is not the case to my knowledge, the authors should town down or remove the related assignments and cited references.

Answer: We thank the reviewer for the insightful comments. We agree with this comment.

(a) We referred to the hot triplet excited state and T_n level in terms of the reversible intersystem crossing mechanism. We did this because there was the possibility of intersystem crossing from the T₂ or T₃ level to the S₁ level based on the small energy gap between the T₂/T₃ and S₁ levels in the calculation data and we could not clearly identify the difference between the T₂ and T₃ levels from the experimental evidence. We accept the reviewer's comment and removed the reference to 'hot triplet state/T_n' or replaced it with 'T₂ level'. We amended the related sentences as below according to the reviewer's comment. If the reviewer feels that further correction is needed, please let us know. We appreciate this important comment.

Revised text)

No.	Position	Before	After
1	Page 2, line 35	To confirm the performance of the novel host materials, we fabricated devices in which the host material was doped with ν -DABNA. The spin-orbit coupling (SOC) of the S ₁ and T ₁ states and that of the S ₁ and T _n states for the materials were calculated and compared.	To confirm the performance of the novel host materials, we fabricated devices in which the host material was doped with ν -DABNA. The spin-orbit coupling (SOC) of the S ₁ and T ₁ states and that of the S ₁ and T ₂ states for the materials were calculated and compared.
2	Page 3, line 38	The calculated ΔE_{ST} values of the newly synthesized host materials were 0.45–0.52 eV (Table S1). In addition, by checking the energy level of the T _n state of these host materials, we confirmed that the energy differences between the T ₂ , T ₃ , and T ₄ levels and the S ₁ level are	The calculated ΔE_{ST} values of the newly synthesized host materials were 0.45–0.52 eV (Table S1). In addition, by checking the energy level of the T ₂ state of these host materials, we confirmed that the energy differences between the T ₂ and the S ₁ level are smaller than the

		smaller than the difference between the T_1 level and the S_1 level (Supplementary Table S2).	difference between the T_1 level and the S_1 level (Supplementary Table S2).
3	Page 4, line 1	On the basis of the reports of Hatakeyama's group, the newly synthesized host materials are expected to exhibit TADF properties and to utilize the hot triplet state	On the basis of the reports of Hatakeyama's group, the newly synthesized host materials are expected to exhibit TADF properties and to utilize the T_2 energy level.
4	Page 4, line 2	In addition, most of the materials have larger SOC values of ($\langle T_n \rangle$) between the S_1 and T_2 levels or between the S_1 level and higher triplet levels such as T_3 or T_4 when compared with the SOC value of ($\langle T_1 \rangle$) between the S_1 and T_1 levels (Supplementary Fig. S2–S8 and Table S2).	In addition, most of the materials have larger SOC values of ($\langle T_2 \rangle$) between the S_1 and T_2 levels compared with the SOC value of ($\langle T_1 \rangle$) between the S_1 and T_1 levels (Supplementary Fig. S2–S8 and Table S2).
5	Page 4, line 6	These results show that the newly synthesized TDBA-based host materials have the advantage of increased luminous efficiency through activation of RISC by simultaneously utilizing the T_n state as well as conventional TADF characteristics.	These results show that the newly synthesized TDBA-based host materials have the advantage of increased luminous efficiency through activation of RISC by simultaneously utilizing the T_2 state as well as conventional TADF characteristics.
6	Page 4, line 30	Although the ΔE_{ST} of S_1-T_1 is relatively large (e.g., >0.3 eV), the T_n state and the S_1 energy gap are relatively small, which is interpreted as promoting RISC from the T_n state to the S_1 level. This result means that the new host materials can promote activation of the RISC process because of the combination of the contribution of the T_n state and the TADF properties.	Although the ΔE_{ST} of S_1-T_1 is relatively large (e.g., >0.3 eV), the T_2 state and the S_1 energy gap are relatively small, which can be interpreted as promoting RISC from the T_2 state to the S_1 level. This result means that the new host materials may promote activation of the RISC process because of the combination of the contribution of the T_2 state and the TADF properties.
7	Page 5, line 31	In a two component system, the newly synthesized host material can facilitate the FRET process to the dopant through RISC activation via the contribution of the T_n state and the contribution of TADF of the T_1 level.	In a two-component system, the newly synthesized host materials may facilitate the FRET process to the dopant through RISC activation of host T_2 level.
8	-	The existence of a T_n state similar to the S_1 energy level might improve the EL performance through the activated RISC process.	This sentence was deleted. And it amended as follow: Second, the enhanced EL performance can be explained by balanced carrier mobility. When the carrier mobility values are compared between TPS-free host materials and TPS-based host materials, the latter shows the relatively superior property to the former, it causes the improved EL performance (Supplementary Table S9).
9	Page 1, line 30	It can also be explained by a hot triplet excited-state contribution	Other factors possibly contributing to the high performance are a T_2 excited-state contribution, high horizontal orientation, and high thermal stability.
10	-	Second, the enhanced EL performance can be explained by the role of the hot triplet excited state	This sentence was deleted. And it amended as follow: Second, the enhanced EL performance can be explained by balanced carrier mobility. When the carrier mobility values are

			compared between TPS-free host materials and TPS-based host materials, the latter shows the relatively superior property to the former, it causes the improved EL performance (Supplementary Table S9).
11	Page 1, line 28	This high performance is attributed to fast energy transfer from the host to the dopant, which is enabled by the external heavy-atom effect of Si, increased spin-orbit coupling, inhibition of aggregation by the bulky tetraphenylsilyl groups, and fast reverse intersystem crossing of the dopant.	This high performance is attributed to fast energy transfer from the host to the dopant, which is enabled by the external heavy-atom effect of Si, inhibition of aggregation by the bulky tetraphenylsilyl groups, and fast reverse intersystem crossing of the dopant.
12	Page 7, line 39	In addition, upon introduction of the TPS moiety, effective energy transfer to the dopant was confirmed by the increased SOC through the external heavy-atom effect.	In addition, upon introduction of the TPS moiety, effective energy transfer to the dopant was confirmed by the external heavy-atom effect.
13	Page 2, line 37	Energy transfer from the new host materials to the dopant was confirmed to occur via the Förster resonance energy transfer (FRET) process through the hot triplet state to the singlet states.	Energy transfer from the new host materials to the dopant was confirmed to occur via the Förster resonance energy transfer (FRET) process through the T ₂ state to the singlet states.

=> In addition, the contents of the T_n state shown in the related DFT calculation results were modified as follows.

Revised Figure and Table in Supporting Information)

Supplementary Fig. S3 | Isosurface of HOMO and LUMO composing S₀→S₁ transition (isovalue = 0.02) with representative electronic transition energies with SOC values of TDBA. TD-B3LYP calculation was conducted at the level of 6-31G(d,p).

Supplementary Fig. S4 | Isosurface of HOMO and LUMO composing $S_0 \rightarrow S_1$ transition (isovalue = 0.02) with representative electronic transition energies with SOC values of TDBA-Ph. TD-B3LYP calculation was conducted at the level of 6-31G(d,p).

Supplementary Fig. S5 | Isosurface of HOMO and LUMO composing $S_0 \rightarrow S_1$ transition (isovalue = 0.02) with representative electronic transition energies with SOC values of mTDBA-Ph. TD-B3LYP calculation was conducted at the level of 6-31G(d,p).

Supplementary Fig. S6 | Isosurface of HOMO and LUMO composing $S_0 \rightarrow S_1$ transition (isovalue = 0.02) with representative electronic transition energies with SOC values of mTDBA-2Ph. TD-B3LYP calculation was conducted at the level of 6-31G(d,p).

Supplementary Fig. S7 | Isosurface of HOMO and LUMO composing $S_0 \rightarrow S_1$ transition (isovalue = 0.02) with representative electronic transition energies with SOC values of TDBA-Si. TD-B3LYP calculation was conducted at the level of 6-31G(d,p).

Supplementary Fig. S8 | Isosurface of HOMO and LUMO composing $S_0 \rightarrow S_1$ transition (isovalue = 0.02) with representative electronic transition energies with SOC values of mTDBA-Si. TD-B3LYP calculation was conducted at the level of 6-31G(d,p).

Supplementary Fig. S9 | Isosurface of HOMO and LUMO composing $S_0 \rightarrow S_1$ transition (isovalue = 0.02) with representative electronic transition energies with SOC values of mTDBA-Si. TD-B3LYP calculation was conducted at the level of 6-31G(d,p).

state	energy (eV)	participating molecular orbitals	transition character
T ₁	2.85	HOMO → LUMO (96%)	$\pi \rightarrow \pi^*$
T ₂	3.31	HOMO-1 → LUMO (58%)	$\pi \rightarrow \pi^*$
S ₁	3.37	HOMO → LUMO (98%)	$\pi \rightarrow \pi^*$

Only molecular orbitals of transition more than 10% were described in the table. Please note that if there are no orbital transitions above 10%, only the three highest orbital transitions are indicated.

Supplementary Fig. S10 | Calculated electronic transitions and each transition character of TDBA.

state	energy (eV)	participating molecular orbitals	transition character
T ₁	2.80	HOMO → LUMO (97%)	$\pi \rightarrow \pi^*$
T ₂	3.16	HOMO-1 → LUMO (62%)	$\pi \rightarrow \pi^*$
S ₁	3.32	HOMO → LUMO (97%)	$\pi \rightarrow \pi^*$

Only molecular orbitals of transition more than 10% were described in the table. Please note that if there are no orbital transitions above 10%, only the three highest orbital transitions are indicated.

Supplementary Fig. S11 | Calculated electronic transitions and each transition character of TDBA-Ph.

state	energy (eV)	participating molecular orbitals	transition character
T ₁	2.75	HOMO → LUMO (96%)	$\pi \rightarrow \pi^*$
T ₂	3.19	HOMO → LUMO+1 (47%), HOMO-1 → LUMO (15%)	$\pi \rightarrow \pi^*$ / partial CT
S ₁	3.23	HOMO → LUMO (98%)	$\pi \rightarrow \pi^*$

Only molecular orbitals of transition more than 10% were described in the table. Please note that if there are no orbital transitions above 10%, only the three highest orbital transitions are indicated.

Supplementary Fig. S12 | Calculated electronic transitions and each transition character of mTDBA-Ph.

state	energy (eV)	participating molecular orbitals	transition character
T ₁	2.67	HOMO → LUMO (96%)	$\pi \rightarrow \pi^*$
T ₂	3.06	HOMO → LUMO+1 (63%)	$\pi \rightarrow \pi^*$
S ₁	3.12	HOMO → LUMO (98%)	$\pi \rightarrow \pi^*$

Only molecular orbitals of transition more than 10% were described in the table. Please note that if there are no orbital transitions above 10%, only the three highest orbital transitions are indicated.

Supplementary Fig. S13 | Calculated electronic transitions and each transition character of mTDDBA-2Ph.

state	energy (eV)	participating molecular orbitals	transition character
T ₁	2.78	HOMO → LUMO (97%)	$\pi \rightarrow \pi^*$
T ₂	3.11	HOMO-1 → LUMO (62%), HOMO-1 → LUMO+1 (12%)	$\pi \rightarrow \pi^*$
S ₁	3.30	HOMO → LUMO (97%)	$\pi \rightarrow \pi^*$

Only molecular orbitals of transition more than 10% were described in the table. Please note that if there are no orbital transitions above 10%, only the three highest orbital transitions are indicated.

Supplementary Fig. S14 | Calculated electronic transitions and each transition character of TDDBA-Si.

state	energy (eV)	participating molecular orbitals	transition character
T ₁	2.76	HOMO → LUMO (96%)	$\pi \rightarrow \pi^*$
T ₂	3.14	HOMO → LUMO+1 (54%)	$\pi \rightarrow \pi^*$ / partial CT
S ₁	3.23	HOMO → LUMO (98%)	$\pi \rightarrow \pi^*$

Only molecular orbitals of transition more than 10% were described in the table. Please note that if there are no orbital transitions above 10%, only the three highest orbital transitions are indicated.

Supplementary Fig. S15 | Calculated electronic transitions and each transition character of mTDBA-Si.

state	energy (eV)	participating molecular orbitals	transition character
T ₁	2.68	HOMO → LUMO (96%)	$\pi \rightarrow \pi^*$
T ₂	3.01	HOMO → LUMO+1 (64%)	$\pi \rightarrow \pi^*$
S ₁	3.13	HOMO → LUMO (98%)	$\pi \rightarrow \pi^*$

Only molecular orbitals of transition more than 10% were described in the table. Please note that if there are no orbital transitions above 10%, only the three highest orbital transitions are indicated.

Supplementary Fig. S16 | Calculated electronic transitions and each transition character of mTDBA-2Si.

Supplementary Table S2 | Summary of Energy state TD-DFT calculation results of TDBA-based material calculated at the B3LYP/6-31G(d,p).

	T ₁ (eV)	T ₂ (eV)	S ₁ (eV)
TDBA	2.85	3.31	3.37
TDBA-Ph	2.80	3.16	3.32
mTDBA-Ph	2.75	3.19	3.23
mTDBA-2Ph	2.67	3.06	3.12
TDBA-Si	2.78	3.11	3.30
mTDBA-Si	2.76	3.14	3.23
mTDBA-2Si	2.68	3.01	3.13

Supplementary Table S3 | Summary of SOC values and representative electronic transition energies of TDBA-based materials.

	$\langle S_1 \hat{H}_{soc} T_1 \rangle$ (cm ⁻¹)	$\langle S_1 \hat{H}_{soc} T_2 \rangle$ (cm ⁻¹)
TDBA	0.01	0.17
TDBA-Ph	0.01	0.17
mTDBA-Ph	0.10	0.44
mTDBA-2Ph	0.13	0.31
TDBA-Si	0.03	0.13
mTDBA-Si	0.10	0.43
mTDBA-2Si	0.13	0.16

(b) On the other hand, regarding the calculated SOC values, recently several important papers have appeared that used small SOC values (<1 cm⁻¹) as outlined below. Thus, we also wanted to report the related SOC values because our materials showed higher SOC values than those in the related references.

	Corresponding author	Position#	Paper#	Name of material	SOC value [cm ⁻¹]
1	Takuji Hatakeyama	Supporting Information Figure S3	Angewandte Chemie International Edition, 2021, 60.33: 17910-17914.	ν -DABNA	0.051
				ν -DABNA-O-Me	0.054
2	Takuji	Main text Figure 2	Journal of the American	ν -DABNA	0.073

	Hatakeyama		Chemical Society, 2021, 144.1: 106-112.	V-DABNA	0.037
3	Chihaya Adachi	Main text Figure 5	Nature communications, 2020, 11.1: 1765.	TMCz-BO	0.124
				TmCz-3P	0.128

Added text) Page 4, line 4

Recently several important studies have examined systems with small SOC values ($< 1 \text{ cm}^{-1}$)^{5,28,29}. The six materials examined in the present work showed higher SOC values than in these previous works.

2. I could not find any excitation spectra throughout the text. In order to exclude any kinetic trace from impurities, the authors should provide excitation spectra of all 6 doped films and check whether they resemble to their absorption spectra.

Answer: We thank the reviewer for this important comment. We added the related excitation spectra to the Supporting Information, as described below. The excitation spectra data of the doped films were matched with the absorption spectra of the constituent non-doped host materials and doped films, as shown below.

Added text) Page 5, line 5

To exclude any kinetic trace effect from impurities, excitation spectra were measured in the doped film state (Supplementary Fig. S20). The excitation spectra data of the doped films were matched with the corresponding absorption spectra of the non-doped host materials and doped films.

Supplementary Fig. S20 | UV-Visible absorption and excitation spectra of 2% v-DABNA films using TDBA-based host materials. a TDBA-Ph. **b** mTDBA-Ph. **c** mTDBA-2Ph. **d** TDBA-Si. **e** mTDBA-Si. **f** mTDBA-2Si.

3. Based on the measured delayed population decays (1~3 μ s), the authors claimed that the RISC of v-DABNA is activated by the newly designed hosts. However, comparisons on the delayed lifetimes between v-DABNA doped in different hosts were not mentioned by the authors. The delayed lifetimes should also be added in Table S5.

Answer: We thank the reviewer very much for this important comment. Accordingly, we added all the related values not only for DOBNA-OAr and mCBP but also for other MR-TADF-type materials, as shown below.

Before text)

The host materials substituted with the TPS moiety showed relatively fast decay, which is advantageous for reducing energy loss via non-radiative channels due to fast energy transfer. This fast decay process, which is approximately four times faster than that recently reported by Hatakeyama's group, means that energy transfer occurs quickly⁶.

After text) Page 5, line 11

The host materials substituted with the TPS moiety showed relatively fast decay, which is advantageous for reducing energy loss via non-radiative channels. This fast decay process, which is faster than the rates of decay recently reported by Prof. Hatakeyama and other groups, means that energy transfer occurs quickly (Supplementary Table S10).

Before)

Supplementary Table S5 | Summary of the reported multiple resonance (MR) and donor-acceptor (DA) type OLEDs (CIE y < 0.15)

Dopant type	Host	Dopant	$E_{QE_{ma}}$	100ni	1,000ni	CIE y	λ [nm]	ref
			x [%]	t [%]	t [%]			
Multiple-resonance type	DOBNA-Tol	v-DABNA-O-Me	29.5	28.8	26.9	0.1	465	1
	DOBNA-Tol	DABNA-NP-TB	19.5	17.5	12	0.11	457	2
	DOBNA-OAr	v-DABNA	34.4	32.8	26	0.11	469	3
	DPEPO	B-O-DPA	16.3	6.5		0.05	443	4
	DBFPO	m-v-DABNA	36.2	-	-	0.12	471	5
	DBFPO	4F-v-DABNA	35.8	-	-	0.08	464	5
	DBFPO	4F-m-v-DABNA	33.7			0.06	461	5
	DBFPO	BN1	31.2	18.3	9.3	0.08	457	6
	DBFPO	BN2	33.2	25.5	15.5	0.11	467	6
DBFPO	BN3	37.6	34	26.2	0.08	458	6	

	mCBP	BBCz-DB	29.3	-	-	0.18	469	7
	mCBP	v-DABNA	23	-	10	0.2	470	8
	mCBP	DABNA-1	13.5			0.09	464	9
	mCBP	DABNA-2	20.2	13.4		0.13	468	9
	mCBP	BOBO-Z	13.6	9.8	3.3	0.04	445	10
	mCBP	BOBS-Z	26.9	24	15	0.06	456	10
	mCBP	BSBS-Z	26.8	24	15.9	0.08	463	10
	mCBP	v-DABNA	24.6	21.2	14.9	0.12	472	10
	mCP:TSPO1	BisICz	6.5			0.04	437	11
	mCP:TSPO1	tBisICz	15.1			0.05	445	11
	mCP:TSPO1	tPBisICz	23.1			0.05	452	11
	mCBP/mCBP -CN	t-DAB-DPA	27.6	21.8	9.2	0.08	459	12
	DBFPO	TDBA-Ac	21.5			0.06	458	13
	DBFPO	PXB-mIC	12.5			0.08	450	14
	DPEPO	OBOtSAc	31.2			0.09 2	452	15
	DPEPO	TDBA-SAF	28.2			0.09	456	16
	DPEPO	CZ-TRZ3	19.2			0.1	450	17
	DPEPO	CZ-TRZ4	18.3			0.09 7	450	17
	DPEPO	ICzAc	13.7			0.09	454	18
	DPEPO	CNICCz	12.4			0.08	449	19
	DPEPO	CNICCz	12.4	6.4		0.08	449	19
	DPEPO	CNICtCz	16	10.7		0.13	456	19
	DPEPO	DtBuAc-DBT	10.5	9.8		0.13	455	20
	DPEPO	DCzBN2	7.7			0.07	417	21
	DPEPO	DCzBN3	10.3			0.06	414	21
	DPEPO	DMACN-B	10			0.04	444	22
	DPEPO	CzBPCN	14			0.12	460	23
Donor-Acceptor type	DPEPO	CzOMeoB	17.3	8.6		0.09	451	24

	DPEPO	CzMeoB	16.1	12.4		0.1	455	24
	DPEPO	DMAC2PTO	15.2			0.11	448	25
	DPEPO	DPFCz-TRZ	15.5	9.7	8.2	0.1	445	26
	DPEPO	DPACpB	12.8	0.7		0.08	457	27
	DPEPO	DPACoOB	16.2			0.13	460	28
	mCP	sAC-sDBB	25.4	20		0.06	444	29
	mCP	sAC-DBB	16.2	11.4		0.07	437	29
	mCP	TB-tCz	15.9			0.06	412	30
	mCP	TB-tPCz	14.1			0.05	420	30
	PPF	OBO-I	21.7	20.6	16.2	0.1	457	31
	PPF	OBO-II	31.7	29	20.8	0.13	464	31
	TDBA-Si	v-DABNA	36.2	35.9	31.3	0.1	465	
This Work	mTDBA-Si	v-DABNA	27.3	26.3	24.1	0.1	465	
	mTDBA-2Si	v-DABNA	38.1	37.7	16.6	0.1	465	

After)

Supplementary Table S10 | Summary of the reported multiple resonance (MR) and donor-acceptor (DA) type OLEDs (CIE y < 0.15)

Dopant type	Host	Dopant	$\tau_{\text{e}}^{\text{a}}$ [μs]	$\text{EQE}_{\text{max}}^{\text{b}}$ [%]	100nit ^b [%]	1,000nit ^b [%]	CIE y ^c	λ [nm] ^d	ref
	DOBNA-Tol	v-DABNA-O-Me	7.7 ^{e)}	29.5	28.8	26.9	0.1	465	1
	DOBNA-Tol	DABNA-NP-TB	90 ^{e)}	19.5	17.5	12	0.11	457	2
	DOBNA-OAr	v-DABNA	4.1 ^{f)}	34.4	32.8	26	0.11	469	3
	DPEPO	B-O-DPA	224 ^{g)}	16.3	6.5		0.05	443	4
	DBFPO	m-v-DABNA	3.09 ^{h)}	36.2	-	-	0.12	471	5
	DBFPO	4F-v-DABNA	3.12 ^{h)}	35.8	-	-	0.08	464	5
	DBFPO	4F-m-v-DABNA	3.19 ^{h)}	33.7			0.06	461	5
Multiple-resonance type	DBFPO	BN1	126.6 ⁱ⁾	31.2	18.3	9.3	0.08	457	6
	DBFPO	BN2	74.6 ⁱ⁾	33.2	25.5	15.5	0.11	467	6
	DBFPO	BN3	17.8 ⁱ⁾	37.6	34	26.2	0.08	458	6
	mCBP	BBCz-DB	86 ^{j)}	29.3	-	-	0.18	469	7
	mCBP	v-DABNA	-	23	-	10	0.2	470	8
	mCBP	DABNA-1	93.7 ^{k)}	13.5			0.09	464	9
	mCBP	DABNA-2	65.3 ^{k)}	20.2	13.4		0.13	468	9
	mCBP	BOBO-Z	7.7 ^{l)}	13.6	9.8	3.3	0.04	445	10

	mCBP	BOBS-Z	7.6 ^{l)}	26.9	24	15	0.06	456	10
	mCBP	BSBS-Z	6.7 ^{l)}	26.8	24	15.9	0.08	463	10
	mCBP	v-DABNA	3.5 ^{l)}	24.6	21.2	14.9	0.12	472	10
	mCP:TSPO1	BisICz	-	6.5			0.04	437	11
	mCP:TSPO1	tBisICz	12.5 ^{m)}	15.1			0.05	445	11
	mCP:TSPO1	tPBisICz	1.74 ^{m)}	23.1			0.05	452	11
	mCBP/mCBP-CN	t-DAB-DPA	22.8 ⁿ⁾	27.6	21.8	9.2	0.08	459	12
	Polymer C	V-DABNA-Mes	2.39 ^{e)}	22.9	20.3	10.9	0.09	480	13
	SBON	v-DABNA	-	27.6	25.7	19.4	0.12	471	14
	SBON-Me	v-DABNA	-	22.8	19.3	13.3	0.12	471	14
	mCBP	v-DABNA	-	17.6	15.6	11.8	0.12	471	14
	DBFPO	TDBA-Ac	1.0 ^{o)}	21.5			0.06	458	15
	DBFPO	PXB-mIC	3.89 ^{p)}	12.5			0.08	450	16
	DPEPO	OBOtSAc	2.92 ^{q)}	31.2			0.092	452	17
	DPEPO	TDBA-SAF	1.34 ^{r)}	28.2			0.09	456	18
	DPEPO	CZ-TRZ3	13.0 ^{s)}	19.2			0.1	450	19
	DPEPO	CZ-TRZ4	10.3 ^{s)}	18.3			0.097	450	19
	DPEPO	ICzAc	9.86 ^{q)}	13.7			0.09	454	20
	DPEPO	CNICCz	6.46 ^{q)}	12.4	6.4		0.08	449	21
	DPEPO	CNICtCz	6.25 ^{q)}	16	10.7		0.13	456	21
	DPEPO	DtBuAc-DBT	136.4 ^{l)}	10.5	9.8		0.13	455	22
	DPEPO	DCzBN2	11.2 ^{q)}	7.7			0.07	417	23
	DPEPO	DCzBN3	13.5 ^{q)}	10.3			0.06	414	23
	DPEPO	DMACN-B	0.77 ^{r)}	10			0.04	444	24
	DPEPO	CzBPCN	48.22 ^{u)}	14			0.12	460	25
	DPEPO	CzOMeOB	52.0 ^{v)}	17.3	8.6		0.09	451	26
	DPEPO	CzMeOB	83.7 ^{v)}	16.1	12.4		0.1	455	26
	mCP	sAC-sDBB	134 ^{w)}	25.4	20		0.06	444	27
	mCP	sAC-DBB	106 ^{w)}	16.2	11.4		0.07	437	27
	mCP	TB-tCz	1.49 ^{w)}	15.9			0.06	412	28
	mCP	TB-tPCz	1.06 ^{w)}	14.1			0.05	420	28
	PPF	OBO-I	1.6 ^{x)}	21.7	20.6	16.2	0.1	457	29
	PPF	OBO-II	1.7 ^{x)}	31.7	29	20.8	0.13	464	29
	TDBA-Si	v-DABNA	0.96	36.2	35.9	31.3	0.1	465	
This Work	mTDBA-Si	v-DABNA	1.93	27.3	26.3	24.1	0.1	465	
	mTDBA-2Si	v-DABNA	0.98	38.1	37.7	16.6	0.1	465	

^{a)}Delayed lifetime calculated by photoluminescence decay. ^{b)}External quantum efficiency at maximum, 100nit, and 1000nit respectively. ^{c)}Commission Internationale de l'Eclairage coordinates from electroluminescence spectrum. ^{d)}Electroluminescence emission maximum. ^{e)}Obtained in 1wt% dopant in PMMA film ^{f)}Obtained in 1wt% dopant in DOBNA-OAr film ^{g)}Obtained in 10wt% dopant in DPEPO film. ^{h)}Obtained in 3wt% dopant in DBFPO film. ⁱ⁾Obtained in 1wt% dopant in DBFPO film. ^{j)}Obtained in 0.01mM toluene solution. ^{k)}Obtained in 1wt% dopant in mCBP film. ^{l)}Obtained in 3wt% dopant in mCBP film. ^{m)}Obtained in 1wt% dopant in mCP:TSPO1 film ⁿ⁾Obtained in 3wt% dopant in mCBP/mCBP-CN film. ^{o)}Obtained in 20wt% dopant in DBFPO film. ^{q)}Obtained in 10wt% dopant in DPEPO film. ^{r)}Obtained in 20wt% dopant in DPEPO

film. ³⁾ Obtained in 6wt% dopant in DPEPO film. ¹⁾ Obtained in 1wt% dopant in xenex film. ⁴⁾ Obtained in 1wt% dopant in polystyrene film. ⁵⁾ Obtained in 0.02mM toluene solution. ⁶⁾ Obtained in 30wt% dopant in mCP film. ⁷⁾ Obtained in 20wt% dopant in PPF film.

4. Continuing suggestion #3, there is surely significant host-guest interaction (either ground state or excited state) that results in such anomalous short delayed lifetimes. For merely v-DABNA dissolved in dilute solution (or other MR-TADF dopants), TADF is usually not feasible due to the high ΔE_{ST} (film: S1-, T1-). However, I cannot see any description on the host-guest interaction or related references.

The authors are encouraged to perform steady state or transient absorption on v-DABNA doped films to capture any plausible ground state complex or exciplex formation. To my knowledge, a typical example was recently published in Nature Photonics 15.10 (2021): 780-786.

Answer: We agree with the reviewer's insightful comment. For a 1 wt% v-DABNA film in DOBNA-OAr (*Nat. Photonics* **13**, 678-682 (2019)) and mCBP (*Nat. Photonics* **15**, 203-207 (2021)), ΔE_{ST} was 0.017 and 0.2 eV, respectively.

The reviewer recommended that we obtain spectroscopic data of the steady state or transient absorption.

Regarding the transient absorption, the related experiments have been started, but the data collection is not finished because of the many kinds of spectroscopy experiments required. Collecting and analyzing the related data in order to clearly understand the mechanism will take us 2 months to 1 year. Also, in the present paper we want to focus only on the material structure affording highly efficient blue EL. Thus, we will prepare another paper including the excitation dynamics data as well as the energy transfer mechanism based on transient absorption spectroscopy. Please understand this situation.

We did, however, measure the steady state absorption and photoluminescence spectra for the 2 wt% and 10 wt% doped films, as shown below. The absorption and photoluminescence spectra were not changed, which means that there is no ground state complex or exciplex formation.

Fig R1. UV-Vis absorption spectra of the synthesized 6 compounds under w/o and w/ 2% dopant.

Fig R2. UV-Vis absorption spectra under the 2% and 10% v-DABNA dopants. (a) TDBA-Ph and (b) TDBA-Si.

Fig R3. PL spectra under the 2% and 10% v-DABNA dopants. (a) TDBA-Ph and (b) TDBA-Si.

Instead, according to the reviewer's comment regarding significant host-guest interaction, we measured the energy transfer values based on the Stern-Volmer equation, as outlined below. The k_q value of each of the three TPS-based host materials was larger than that of the corresponding TPS-free host material. We added the related description to the text and Supporting Information, as shown below.

Added text) Page 5, line 33

To confirm that efficient energy transfer occurs between the TPS-based host materials and v-DABNA, we measured the energy transfer values of the Stern-Volmer equation (Supplementary Table S7). The k_q values of the three TPS-based host materials are larger than those of the corresponding TPS-free host materials.

Supplementary Table S7 | Rate constant of energy transfer between host and dopant based on Stern-Volmer equation.

	TDBA-Ph	mTDBA-Ph	mTDBA-2Ph	TDBA-Si	mTDBA-Si	mTDBA-2Si
Slope	0.13	0.048	0.14	0.43	0.19	0.16
k_q^a	3.43×10^7	1.02×10^7	3.23×10^7	9.48×10^7	4.08×10^7	5.90×10^7

^a Energy transfer that occurs between host and dopant: $I_0/I = 1 + k_{\text{quenching}} \times [A]$.

5. Continuing suggestion #4, the band alignment between host and dopant materials will be the determining factor for the device performance if host-guest interactions exist. Hence, the authors should experimentally access the HOMO and LUMO energy with photoelectron spectroscopy or cyclic voltammetry.

Answer: We thank the reviewer for this comment. The HOMO absolute value was determined by photoelectron spectroscopy (AC-2) and the LUMO value was derived from the optical band gap, as shown in Table 1. We added the photoelectron spectroscopy (AC-2) data to the Supporting Information as shown below.

Before)

The highest occupied molecular orbital (HOMO) energy levels were determined with ultraviolet photoelectron yield spectroscopy (Riken Keiki AC-2).

After) Page 8, line 18

The highest occupied molecular orbital (HOMO) absolute energy levels were determined by photoelectron yield spectroscopy (Riken Keiki AC-2). (Supplementary Fig. S18).

Supplementary Fig. S18 | Photoelectron spectroscopy (AC-2) of TDBA-based host materials. a TDBA. **b** TDBA-Ph. **c** mTDBA-Ph. **d** mTDBA-2Ph. **e** TDBA-Si. **f** mTDBA-Si. **g** mTDBA-2Si.

6. The scheme in Figure 2c is incorrect considering from a kinetic point of view. To initiate, when monitoring at the wavelength of dopant's emission, we could observe a tri-exponential decay as seen in Figure 2b. If the authors successfully derive the mathematical relations for this type of system, one will find that the energy transfer rate actually refers to the second lifetime but not the third (longest) lifetime. Additionally, the third lifetime corresponds to the population decay of dopant's triplet state. Therefore, the energy transfer rate for the silicon based hosts are not faster than that of phenyl-based hosts. Figure 2a also showed an emission shoulder for mTDBA-2Si, which indicates its inferior transfer efficiency comparing to other hosts.

Answer: We thank the reviewer for the insightful comments.

According to the reviewer's comment, we divided the exponential decay into 3 components and derived 3 decay lifetimes, as shown below. Comparing the second exponential decay values (τ_2) of the TPS-based hosts with those of the TPS-free hosts, the silicon-based hosts TDBA-Si and mTDBA-Si showed slightly faster decay than TDBA-Ph and mTDBA-Ph, respectively, whereas mTDBA-2Si exhibited slightly slower decay than mTDBA-2Ph (Table S5). We added the related values to Table S5 without a related description in the text.

Before)

When TDBA-Ph, mTDBA-Ph, mTDBA-2Ph, TDBA-Si, mTDBA-Si, and mTDBA-2Si were used as hosts, the decay times were 2.99, 2.76, 1.86, 0.96, 1.93, and 0.98 μ s, respectively.

After text) Page 5, line 10

When TDBA-Ph, mTDBA-Ph, mTDBA-2Ph, TDBA-Si, mTDBA-Si, and mTDBA-2Si were used as hosts in doped films, the average decay times were 2.99, 2.76, 1.86, 0.96, 1.93, and 0.98 μs , respectively (Table 2). The host materials substituted with the TPS moiety showed relatively fast decay, which is advantageous for reducing energy loss via non-radiative channels. This fast decay process, which is faster than the rates of decay recently reported by Prof. Hatakeyama and other groups, means that energy transfer occurs quickly (Supplementary Table S10). As a result of the transient PL measurement in the doped films state, the exciton decay behavior can be divided into three exponential decay components (Supplementary Table S5). The energy transfer rate from host material to dopant corresponds to the second lifetime (τ_2) and the population decay of the dopant's triplet state corresponds to the third lifetime (τ_3). Comparing the second exponential decay values (τ_2) of the TPS-based hosts with TPS-free hosts, TDBA-Si and mTDBA-Si showed slightly faster decay than TDBA-Ph and mTDBA-Ph, respectively.

Supplementary Table S5 | Fitting the decay curves triexponentially according to host materials in doped films.

Materials	τ_1 (μs)	τ_2 (μs)	τ_3 (μs)
TDBA-Ph	0.01	0.279	2.318
mTDBA-Ph	0.024	0.261	1.821
mTDBA-2Ph	0.015	0.228	1.250
TDBA-Si	0.010	0.257	1.05
mTDBA-Si	0.022	0.25	2.00
mTDBA-2Si	0.014	0.260	1.00

(3) We agree with the reviewer's comment regarding the inferior transfer efficiency of mTDBA-2Si because of the residual host emission of mTDBA-2Si in the doped film PL spectrum. This observation might be explained by the relatively slow energy transfer in the second exponential decay value, as noted by the reviewer.

Added text) Page 5, line 2

The PL spectrum of the doped mTDBA-2Si film showed only the residual host emission shoulder peak, indicating imperfect energy transfer from host to dopant.

7. Continuing suggestion #6, equation (1)~(7) are used to access kinetic parameters when TADF is achieved by the dopant molecule itself. As the authors showed, an energy transfer pathway also involves in the excited state dynamics. As a result, all kinetic parameters deduced in this manuscript have to be re-evaluated. The authors should simulate the kinetic parameters again in consideration of the energy transfer.

Answer: We thank the reviewer for this important comment. According to the reviewer's comment, we calculated $K_{\text{FRET}} (= k_{\text{PF}} - k_{\text{r,s}} - k_{\text{ISC}})$ values based on the paper of Prof. Kwon (Adv. Funct. Mater. 2021, 31, p2105805), as shown in Table S8.

Two of the TPS-based host materials, mTDBA-Si and mTDBA-2Si, showed larger k_{FRET} values than the corresponding TPS-free host materials. TDBA-Si, by contrast, showed a lower k_{FRET} value than TDBA-Ph. We could not identify a clear reason for the observed k_{FRET} values. We added a related description to the text and Supporting Information, as shown below.

Added text) Page 5, line 35

Also, calculation of the k_{FRET} values confirmed that two materials containing the TPS moiety (mTDBA-Si and mTDBA-2Si) show faster k_{FRET} than the TPS-free materials (Supplementary Table S8)³⁵.

Supplementary Table S8 | FRET calculation summary between newly synthesized host materials and v-DABNA.

	PLQY	$J (\lambda)^a$ [mol ⁻¹ dm ³ cm ⁻¹ nm ⁴]	R_F^b [nm]	R^c [nm]	k_{FRET}^d [10 ⁸ /s]
TDBA-Ph	0.82	9.69 x 10 ¹⁴	2.99	2.84	1.47
mTDBA-Ph	0.93	9.55 x 10 ¹⁴	2.98	2.82	1.38
mTDBA-2Ph	0.86	1.00 x 10 ¹⁵	3.01	2.93	1.32
TDBA-Si	0.82	1.02 x 10 ¹⁵	3.02	3.11	1.07
mTDBA-Si	0.94	9.16 x 10 ¹⁴	2.96	2.74	1.44
mTDBA-2Si	0.92	9.79 x 10 ¹⁴	2.99	2.84	2.08

^a Spectral overlap between PL emission of TDBA-based host materials and absorption spectrum of v-DABNA. ^b FRET radius. ^c Intermolecular distance. ^d FRET rate constant: $k_{\text{FRET}} = k_{\text{PF}} - k_{\text{r,s}} - k_{\text{ISC}}$

Supplementary Table S4 | Rate constant for TDBA based host materials (non-doped film) at room temperature.

	TDBA-Ph	mTDBA-Ph	mTDBA-2Ph	TDBA-Si	mTDBA-Si	mTDBA-2Si
Φ	0.62	0.65	0.63	0.66	0.61	0.75
Φ_F	0.538	0.589	0.603	0.644	0.592	0.731
Φ_{TADF}	0.082	0.061	0.027	0.016	0.018	0.019

τ (ns)	9.35	10.01	8.92	7.80	11.1	6.50
τ_{TADF} (μs)	0.91	1.25	2.33	3.53	6.65	6.49
k_{F} ($\times 10^7$)	5.75	5.87	6.76	8.25	5.32	11.2
k_{IC} ($\times 10^7$)	3.53	3.16	3.97	4.25	3.40	3.75
k_{ISC} ($\times 10^7$)	10.4	9.27	4.85	3.14	2.63	3.92
Φ_{IC}	0.33	0.317	0.354	0.332	0.379	0.244
Φ_{ISC}	0.13	0.09	0.04	0.02	0.03	0.03
k_{TADF} ($\times 10^5$)	6.82	5.22	2.71	1.87	9.18	1.16
k_{RISC} ($\times 10^5$)	4.23	3.39	1.71	1.23	5.60	8.66

Additionally, we measured the energy transfer values based on the Stern-Volmer equation as outlined below. The k_{q} values of the three TPS-based host materials were larger than those of the corresponding TPS-free host materials. We added a related description to the text and Supporting Information as outlined below.

Added text) Page 5, line 33

To confirm that efficient energy transfer occurs between the TPS-based host materials and ν -DABNA, we measured the energy transfer values of the Stern-Volmer equation (Supplementary Table S7). The k_{q} values of the three TPS-based host materials are larger than those of the corresponding TPS-free host materials. Also, calculation of the k_{FRET} values confirmed that two materials containing the TPS moiety (mTDBA-Si and mTDBA-2Si) show faster k_{FRET} than the TPS-free materials (Supplementary Table S8)³⁵.

Supplementary Table S7 | Rate constant of energy transfer between host and dopant based on Stern-Volmer equation.

	TDBA-Ph	mTDBA-Ph	mTDBA-2Ph	TDBA-Si	mTDBA-Si	mTDBA-2Si
Slope	0.13	0.048	0.14	0.43	0.19	0.16
k_{q}^{a}	3.43×10^7	1.02×10^7	3.23×10^7	9.48×10^7	4.08×10^7	5.90×10^7

^a Energy transfer that occurs between host and dopant: $I_0/I = 1 + k_{\text{quenching}} \times [A]$

8. How did the authors confirm that the FRET takes place from host's hot triplet state to dopant's singlet state? This assignment is fundamentally weird! The authors are recommended to show the spectral overlap between the fluorescence spectrum of host and the absorption spectrum of dopant in the supporting information.

Answer: We thank the reviewer for this important comment. Regarding the FRET from the host's hot triplet state to the dopant's singlet state, there was an error in the related text in the original submission. We apologize for this. From the T₂ level, it first moves to the S₁ level of the host through RISC, and this energy is transferred to the singlet state level of the dopant. We added the related spectral overlap figures of the host and dopant to the Supporting Information, and amended the related text as shown below.

Before)

In a two component system, the newly synthesized host material can facilitate the FRET process to the dopant through RISC activation via the contribution of the T_n state and the contribution of TADF of the T₁ level.

After) Page 5, line 35

In a two component system, the newly synthesized host materials may facilitate the FRET process to the dopant through RISC activation of host T₂ level.

Added text) Page 5, line 7

This comparison revealed overlap between the PL spectra of the TDBA-based host materials in the neat film state and the UV absorption spectra of the dopant in the solution state, indicating that energy transfer can readily occur (Supplementary Fig. S21).

Supplementary Fig. S21 | Spectral overlap between v-DABNA and TDBA-based host. a TDBA-Ph.

b mTDBA-Ph. **c** mTDBA-2Ph. **d** TDBA-Si. **e** mTDBA-Si. **f** mTDBA-2Si.

9. In Table S5, the EQE roll off for mTDBA-2Si is much more severe than that of TDBA-Si and mTDBA-Si. Please explain.

Answer: We thank the reviewer for this important comment. The reason why mTDBA-2Si exhibits more severe roll-off than the other two materials can be explained as follows. The values of the charge balance, which is the hole mobility of the hole-only device (HOD) divided by the electron mobility of the electron-only device (EOD), for the doped devices with TDBA-Si, mTDBA-Si, and mTDBA-2Si were 1.05, 1.29, and 4.00, respectively. This indicates that mTDBA-2Si has an imbalance in the carrier mobility of holes and electrons, which can account for the severe roll-off in EQE observed for this host material.

Added text) Page 7, line 10

The more severe roll-off observed for mTDBA-2Si compared to TDBA-Si and mTDBA-Si may be due to the higher imbalance in charge mobility of holes and electrons of mTDBA-2Si (Supplementary Table S9). Specifically, the charge balance value, which is the hole mobility of the hole-only device (HOD) divided by the electron mobility of the electron-only device (EOD) in the doped devices, of TDBA-Si, mTDBA-Si, and mTDBA-2Si was 1.05, 1.29, and 4.00, respectively. The imbalanced carrier mobility of holes and electrons in mTDBA-2Si can cause severe EQE roll-off in device performance.

Supplementary Table S9 | Charge mobility of TDBA-based host materials.

Compounds	Mobility (cm ² /Vs) @ 1V			
	Non-doped film		Doped film	
	HOD	EOD	HOD	EOD
TDBA-Ph	1.59 x 10 ⁻⁵	1.01 x 10 ⁻⁶	8.71 x 10 ⁻⁷	7.79 x 10 ⁻⁸
mTDBA-Ph	1.34 x 10 ⁻⁵	9.61 x 10 ⁻⁷	5.51 x 10 ⁻⁷	6.51 x 10 ⁻⁸
mTDBA-2Ph	2.67 x 10 ⁻⁵	8.89 x 10 ⁻⁷	1.45 x 10 ⁻⁶	6.96 x 10 ⁻⁸
TDBA-Si	5.02 x 10 ⁻⁷	7.04 x 10 ⁻⁷	3.18 x 10 ⁻⁸	3.04 x 10 ⁻⁸
mTDBA-Si	2.83 x 10 ⁻⁷	7.03 x 10 ⁻⁷	2.39 x 10 ⁻⁸	1.85 x 10 ⁻⁸
mTDBA-2Si	9.40 x 10 ⁻⁷	4.58 x 10 ⁻⁷	5.63 x 10 ⁻⁸	1.41 x 10 ⁻⁸

Minor revisions:

1. The reported Δ ESTs were calculated by subtracting the S1 energy with T1 energy, which the authors obtained by subtracting the spectral onset of room temperature fluorescence with that of 77 K phosphorescence. Alternatively, the authors are recommended to calculate the Δ ESTs by subtracting the spectral onset of 77 K fluorescence with that of 77 K phosphorescence.

Answer: We appreciate this important comment. Accordingly, we measured the spectra of 77 K fluorescence and the related data, as shown below.

Before)

Supplementary Fig. S17 | Absorption (dash), room temperature photoluminescence (line), and low temperature photoluminescence (dot) at 77K spectra of neat film. a TDBA-Ph. b mTDBA-Ph. c mTDBA-2Ph. d TDBA-Si. e mTDBA-Si. f mTDBA-2Si.

After)

Supplementary Fig. S19 | Absorption spectra(dash) at room temperature, low temperature photoluminescence without (line) and with delay (dot) at 77K spectra of neat film. a TDDBA-Ph. b mTDDBA-Ph. c mTDDBA-2Ph. d TDDBA-Si. e mTDDBA-Si. f mTDDBA-2Si.

Before)

Table 1 | Summary of the photophysical properties of the TDDBA-based materials.

	Solution			Neat Film								
	λ_{ab}^a (nm)	λ_{em}^a (nm)	FWH M (nm)	λ_{ab}^a (nm)	λ_{em}^a (nm)	FWH M (nm)	PLQ Y (%)	$E_s / E_T / \Delta E_{ST}^b$ (eV)	τ_d (μs) ^c	HOM O ^d (eV)	LUM O ^d (eV)	E_g^d (eV)
TDDBA	383	403	27	—	—	—	—	3.15/2.81/0.34	—	6.02	2.87	3.15
TDDBA-Ph	388	408	27	392	418	66	62	3.10/2.79/0.31	0.91	5.75	2.65	3.10
mTDDBA-Ph	392	421	34	398	434	75	65	3.00/2.70/0.30	1.25	5.68	2.64	3.04
mTDDBA-2Ph	400	433	37	406	450	70	63	3.13/2.76/0.37	2.33	5.64	2.67	2.97
TDDBA-Si	389	411	27	393	425	73	66	3.10/2.73/0.37	3.53	5.73	2.64	3.09
mTDDBA-Si	393	421	33	398	449	79	61	2.99/2.70/0.29	6.65	5.71	2.66	3.05
mTDDBA-2Si	401	432	35	405	447	75	75	3.15/2.81/0.34	6.49	5.65	2.69	2.96

After)

Table 1 | Summary of the photophysical properties of the TDBA-based materials.

	Solution			Neat Film								
	λ_{ab}^a (nm)	λ_{em}^a (nm)	FWH M (nm)	λ_{ab}^a (nm)	λ_{em}^a (nm)	FWH M (nm)	PLQ Y (%)	$E_S / E_T / \Delta E_{ST}^b$ (eV)	τ_d (μs) ^c	HOM O ^d (eV)	LUM O ^d (eV)	E_g^d (eV)
TDBA	383	403	27	—	—	—	—	—/—/—	—	6.02	2.87	3.15
TDBA-Ph	388	408	27	392	418	66	62	3.06/ 2.82/ 0.24	0.91	5.75	2.65	3.10
mTDBA-Ph	392	421	34	398	434	75	65	2.99/ 2.78/ 0.21	1.25	5.68	2.64	3.04
mTDBA-2Ph	400	433	37	406	450	70	63	2.91/ 2.71/ 0.20	2.33	5.64	2.67	2.97
TDBA-Si	389	411	27	393	425	73	66	3.02/ 2.77/ 0.25	3.53	5.73	2.64	3.09
mTDBA-Si	393	421	33	398	449	79	61	3.01/ 2.73/ 0.28	6.65	5.71	2.66	3.05
mTDBA-2Si	401	432	35	405	447	75	75	2.96/ 2.72/ 0.24	6.49	5.65	2.69	2.96

Before)

When a phenyl or TPS group was substituted at the same position in TDBA, similar S_1 and T_1 levels were observed and the ΔE_{ST} was approximately 0.29–0.37 eV for all the materials (Table 1).

After) Page X, line Y

When a phenyl or TPS group was substituted at the same position in TDBA, similar S_1 and T_1 levels were observed and the ΔE_{ST} was approximately 0.20–0.28 eV for all the materials (Table 1).

2. In the Method part, there is a typo in Equation 5. It should be $\Phi_{ISC} = 1 - \Phi_F - \Phi_{IC}$.

Answer: We apologize for this typo, which we have corrected.

3. The authors claimed that aggregations were avoided by inserting the silicon moiety. Accordingly, the packing environment as well as the intermolecular distances should be provided in a figure.

Answer: According to the reviewer's comment, we added the related data and an explanation regarding the intermolecular distances as well as the XRD data, as outlined below.

Added text) Page 3, line 13

Single crystal analysis of TDBA-Ph and TDBA-Si confirmed that TDBA-Si containing the TPS moiety had a longer plane-to-plane distance than did TDBA-Ph (Supplementary Fig. S1).

	Distance of plant to plane
TDBA-Ph	3.426 Å
TDBA-Si	4.197 Å

Supplementary Fig. S1 | Single crystal XRD analysis. a TDBA-Ph. b TDBA-Si.

4. Please unify the format of Reference section.

Answer: We thank the reviewer for this comment. We have unified the referencing format.

Reviewer #2 (Remarks to the Author):

In this paper, the authors reported three MR-TADF molecules by using the silicon atom. The devices fabricated by using them as host materials showed excellent performance with the EQEs up to 36.2%. But, I think the primary reason of achieving excellent performance is caused by the high device technology level and by using the star MR-TADF emitter, v-DABNA, which has been explored widely as the emitter for high-performance OLEDs with the EQE exceeds 30%. Moreover, the molecular design is not novel (*Angew. Chem. Int. Ed.* 2015, 54, 13581 –13585, Hatakeyama) and the mechanism of the energy transfer proposed in this paper has no definitive proof. Therefore, I do not think that this paper can be published in *Nature Communications*. Here are some suggestions for authors, hoping to improve the quality of the manuscript and then published on a more specialized journal in the future.

Issue 1 - answer:

We thank the reviewer for these comments. We agree that the v-DABNA emitter plays a very important role in achieving high efficiency. However, to our knowledge only one study using the v-DABNA emitter by Prof. Adachi's group has achieved an EQE higher than we achieved in this study, but that study (which reported an EQE of 41%) used a three-component system including host, sensitizer, and dopant (*Nat. Photonics.*, 15, 203-205 (2021)). We report an EQE 38% using a two-component conventional emitter system including only host and dopant, with the high EQE deriving from the inclusion of Si atoms in the host material. Moreover, unlike the device of Prof. Adachi's group, our devices based on the Si-containing host material exhibited low roll-off that yields an EQE of 30% or more even at 1,000 nit. Among other MR-TADF emitters including v-DABNA, there is no study that achieves an EQE of 30% at 1,000 nit. Please refer to Supplementary Table S10.

Issue 2 – answer (Novelty):

The paper mentioned by the reviewer (*Angew. Chem. Int. Ed.* 2015, 54, 13581 –13585, Hatakeyama) has the main structures as shown in the figure below.

Fig. R1 Molecular structures of DOBNA type host materials (*Angew. Chem. Int. Ed.* 2015, 54, 13581-13585).

Among the seven host materials we proposed (Fig. R2), three reference materials (TDBA-Ph, mTDBA-Ph, mTDBA-2Ph) are similar to the chemical structures of the materials in the paper mentioned by the reviewer. However, the three main materials in our study (TDBA-Si, mTDBA-Si, mTDBA-2Si) have completely different chemical structures. The related explanation was not included in our original submission due to the limitation of the number of pages. We apologize if this caused confusion. The following contents are included in the Introduction and supporting parts of the revised manuscript.

Fig. R2 Molecular structures of the host material proposed in this study.

Added text) Page 2, line 32 in Introduction

The novelty of the structural design concept of the host material for our proposed MR-TADF emitter is the inclusion of silicon atoms. These atoms have two functions: they bring into play the external heavy atom effect and they increase the intermolecular distance in the host material.

Distance of plant to plane	
TDBA-Ph	3.426 Å
TDBA-Si	4.197 Å

Supplementary Fig. S1 | Single crystal XRD analysis. a TDBA-Ph. b TDBA-Si.

Added text) Page 3, line 13

Single crystal analysis of TDBA-Ph and TDBA-Si confirmed that TDBA-Si containing the TPS moiety had a longer plane-to-plane distance than did TDBA-Ph (Supplementary Fig. S1).

Added text Fig. S1 in Supporting Information)

The novelty of the structural design concept of the host material for our proposed MR-TADF emitter is to include silicon atoms. TPS-based host materials have the effect of reducing intermolecular interactions. Since the C-Si bond length is relatively longer than the C-C bond length, intermolecular packing can be prevented. Single crystal XRD of two materials, TDBA-Ph and TDBA-Si, was analyzed to confirm the intermolecular distance in the symmetrical molecular structure. As shown in Supplementary Fig. S1, it can be confirmed that the distance between planes of each molecule is longer for TDBA-Si molecule containing Si atoms compared to TDBA-Ph.

Issue 3 – answer (The mechanism of the energy transfer):

We did not provide the related mechanism in the first submission. We apologize for this. The mechanism proof for the energy transfer was confirmed through not only the result of the Stern-Volmer experiment but also the second exponential decay value in the transient PL decay of the doped films. Regarding the results of the Stern-Volmer experiment, the corresponding answer is described below in our response reviewer #2-1. Also, the following data and text are included in the revised manuscript.

Before)

When TDBA-Ph, mTDBA-Ph, mTDBA-2Ph, TDBA-Si, mTDBA-Si, and mTDBA-2Si were used as hosts, the decay times were 2.99, 2.76, 1.86, 0.96, 1.93, and 0.98 μs , respectively.

After text) Page 5, line 10

When TDBA-Ph, mTDBA-Ph, mTDBA-2Ph, TDBA-Si, mTDBA-Si, and mTDBA-2Si were used as hosts in doped films, the average decay times were 2.99, 2.76, 1.86, 0.96, 1.93, and 0.98 μs , respectively (Table 2). The host materials substituted with the TPS moiety showed relatively fast decay, which is advantageous for reducing energy loss via non-radiative channels. This fast decay process, which is faster than the rates of decay recently reported by Prof. Hatakeyama and other groups, means that energy transfer occurs quickly (Supplementary Table S10). As a result of the transient PL measurement in the doped films state, the exciton decay behavior can be divided into three exponential decay components (Supplementary Table S5). The energy transfer rate from host material to dopant corresponds to the second lifetime (τ_2) and the population decay of the dopant's triplet state corresponds to the third lifetime (τ_3). Comparing the second exponential decay values (τ_2) of the TPS-based hosts with TPS-free hosts, TDBA-Si and mTDBA-Si showed slightly faster decay than TDBA-Ph and mTDBA-Ph, respectively.

Supplementary Table S5 | Fitting the decay curves triexponentially according to host materials in doped films.

Materials	τ_1 (μs)	τ_2 (μs)	τ_3 (μs)
TDBA-Ph	0.01	0.279	2.318

mTDBA-Ph	0.024	0.261	1.821
mTDBA-2Ph	0.015	0.228	1.250
TDBA-Si	0.010	0.257	1.05
mTDBA-Si	0.022	0.25	2.00
mTDBA-2Si	0.014	0.260	1.00

1. The authors should provide more comprehensive sets of experiments and theoretical calculations supporting the fast energy transfer from the host materials containing the silicon atom to dopant.

Answer: We thank the reviewer for this insightful comment. We added the related experimental data and explanation, including the Stern-Volmer equation and second exponential decay time in the transient PL spectrum, as outlined below.

(1) Stern-Volmer equation (energy transfer)

Added) Page 5, line 33

To confirm that efficient energy transfer occurs between the TPS-based host materials and ν -DABNA, we measured the energy transfer values of the Stern-Volmer equation (Supplementary Table S7). The k_q values of the three TPS-based host materials are larger than those of the corresponding TPS-free host materials.

Supplementary Table S7 | Rate constant of energy transfer between host and dopant based on Stern-Volmer equation.

	TDBA-Ph	mTDBA-Ph	mTDBA-2Ph	TDBA-Si	mTDBA-Si	mTDBA-2Si
Slope	0.13	0.048	0.14	0.43	0.19	0.16
k_q^a	3.43×10^7	1.02×10^7	3.23×10^7	9.48×10^7	4.08×10^7	5.90×10^7

^a Energy transfer that occurs between host and dopant: $I_0/I = 1 + k_{\text{quenching}} \times [A]$

(2) Analysis of exciton behavior through transient PL in doped state.

Before)

When TDBA-Ph, mTDBA-Ph, mTDBA-2Ph, TDBA-Si, mTDBA-Si, and mTDBA-2Si were used as hosts, the decay times were 2.99, 2.76, 1.86, 0.96, 1.93, and 0.98 μs , respectively.

After text) Page 5, line 10

When TDBA-Ph, mTDBA-Ph, mTDBA-2Ph, TDBA-Si, mTDBA-Si, and mTDBA-2Si were used as hosts in doped films, the average decay times were 2.99, 2.76, 1.86, 0.96, 1.93, and 0.98 μs , respectively (Table 2). The host materials substituted with the TPS moiety showed relatively fast decay, which is advantageous for reducing energy loss via non-radiative channels. This fast decay process, which is

faster than the rates of decay recently reported by Prof. Hatakeyama and other groups, means that energy transfer occurs quickly (Supplementary Table S10). As a result of the transient PL measurement in the doped films state, the exciton decay behavior can be divided into three exponential decay components (Supplementary Table S5). The energy transfer rate from host material to dopant corresponds to the second lifetime (τ_2) and the population decay of the dopant's triplet state corresponds to the third lifetime (τ_3). Comparing the second exponential decay values (τ_2) of the TPS-based hosts with TPS-free hosts, TDBA-Si and mTDBA-Si showed slightly faster decay than TDBA-Ph and mTDBA-Ph, respectively.

Supplementary Table S5 | Fitting the decay curves triexponentially according to host materials in doped films.

Materials	τ_1 (μs)	τ_2 (μs)	τ_3 (μs)
TDBA-Ph	0.01	0.279	2.318
mTDBA-Ph	0.024	0.261	1.821
mTDBA-2Ph	0.015	0.228	1.250
TDBA-Si	0.010	0.257	1.05
mTDBA-Si	0.022	0.25	2.00
mTDBA-2Si	0.014	0.260	1.00

2. For the mechanism of the energy transfer in fig 2c, here is a certain possibility of the Dexter energy transfer from host to dopant because the RISC rate of the host materials is not fast enough.

Answer: We thank the reviewer for this extremely important comment. Indeed, we had thought of the possibility of Dexter energy transfer from host to dopant. As shown in the data below, we performed related experiments using the phosphorescent sky-blue dopant FirPic with TDBA-Ph, TDBA-Si, and DMAC-DPS as the host material. DMAC-DPS is the conventional TADF host based on a Donor-Acceptor type (Org. Electron. 2017, 41, 237). The device configuration was ITO/NPB (40 nm)/TCTA (15 nm)/mCP (15 nm)/ TDBA-Ph or TDBA-Si or DMAC-DPS: 10% FirPic (20 nm)/TmPyPB (40 nm)/LiF/Al.

Fig XX. Molecule structure of DMAC-DPS (TADF material).

As shown in the table below, the conventional host material DMAC-DPS showed a maximum EQE of 4.38%, whereas TDBA-Ph and TDBA-Si exhibited maximum EQEs of 8.67% and 15.4%, respectively. These findings indicate the strong possibility of Dexter energy transfer. However, we have not acquired all the related detailed data because we need to try to obtain data on the three primary colors for all devices. We will publish another paper with the results of the phosphorescence experiments. If the reviewer strongly recommends the inclusion of this data, we will add it into the current manuscript.

Fig R1. EL performances of Ph-OLEDs using TDBA-Ph, TDBA-Si, and DMAC-DPS. a) Current efficiency (CE)-L curves and b) EL spectra.

Table R2. EL performances of Ph-OLEDs using TDBA-Ph, TDBA-Si and DMAC-DPS as host materials.

EMLs	Max. L (cd/m ²)	Current efficiency (cd/A)			EQE (%)			CIE (x, y)	EL _{max} (nm)
		Max.	500 nit	1,000 nit	Max.	500 nit	1,000 nit		
TDBA-Ph: 10% FirPic	6,700	17.5	17.5	16.8	8.67	8.67	8.26	0.173, 0.339	469, 496
TDBA-Si: 10% FirPic	7,200	35.3	32.1	31.3	15.4	14.1	13.8	0.173, 0.339	469, 496
DMAC-DPS: 10% FirPic	2,000	9.61	8.85	6.50	4.38	4.03	2.95	0.173, 0.339	469, 496

Issue: Moreover, fig 2c shows the mechanism of the energy transfer under the electro-excitation, not for the photo-excitation, so it is not suitable in fig 2, but should be in fig 3 instead.

Answer: We thank the reviewer for this comment. We changed the location of the related figure as shown below.

Fig. 3a shows a proposed energy-transfer mechanism between the host and the ν -DABNA dopant used in this study.

Before)

Fig. 2 | Photophysical properties and schematic of the 2 wt% ν -DABNA-doped TDBA-based films.
a Photoluminescence spectra. **b** Transient decay spectra of the doped films. **c** Schematic of the conceived energy-transfer mechanism for the host and the ν -DABNA dopant.

After)

Fig. 3 | a Schematic of the conceived energy-transfer mechanism for the host and the ν -DABNA dopant. **b** Energy-level diagram of doped OLED devices. **c** Molecular structures used in each layer. **d** Current efficiency (CE)–luminance (L) curves. **e** External quantum efficiency (EQE)–L curves. **f** EL spectra (inset: OLED driving image at 7 V).

3. I do not think the silicon is a heavy atom, so here there is no strong evidence of having heavy-atom effect. The values of the SOC matrix elements of the molecules in this paper are too small, which are not comparable with the ones for those molecules containing the sulfur or selenium atom (Angew. Chem. Int. Ed. 2022, e202205684; Nat. Photonics 2022, 16, 803–810, Chuluo Yang).

Answer: We agree with the reviewer’s comment regarding the relatively small SOC values compared with those mentioned by the reviewer. However, recently several important papers have been published that used small SOC values of less than 1 cm^{-1} (see below) to explain differences in EL efficiency. We also want to use the SOC value in order to understand EL efficiency because our materials showed relatively high SOC values compared to those of the references below. Additionally, the reference paper mentioned by the reviewer used the words ‘heavy atom’ when referring to sulfur and selenium atoms.

	Corresponding author	paper#	paper#	Name of material	SOC value [cm ⁻¹]
1	Takuji Hatakeyama	Supporting Information Figure S3	Angewandte Chemie International Edition, 2021, 60.33: 17910-17914.	v-DABNA	0.051
				v-DABNA-O-Me	0.054
2	Takuji Hatakeyama	Main text Figure 2	Journal of the American Chemical Society, 2021, 144.1: 106-112.	v-DABNA	0.073
				V-DABNA	0.037
3	Chihaya Adachi	Main text Figure 5	Nature communications, 2020, 11.1: 1765.	TmCz-BO	0.124
				TmCz-3P	0.128

Reviewer #3 (Remarks to the Author):

The manuscript by Jongwook Park et al. reports TDBA-based host materials possessing one or two tetraphenylsilyl (TPS) groups. The devices using the host materials and v-DABNA exhibited high EQE (up to 36.2%) and small efficiency roll-off (31.3% at 1000 cd/m²). This is one of the best performances of blue MR-TADF-OLEDs, as summarized in Figure 4. However, the claims and discussion regarding the heavy atom effect of the TPS group are not well supported by the experimental and computational results. It should be re-evaluated after significant and careful revisions. Detailed comments on the authors' claims are listed below.

1. Authors' claim: Possible explanations for the enhanced EL performance of the TPS-containing host materials compared with that of the phenyl-substituted host materials are as follows. First, the enhanced EL performance can be explained by the rapid energy transfer of the three TPS-substituted materials to the dopant. In the v-DABNA-doped films of the three materials with the TPS moiety, the delayed lifetime was as much as three times shorter than that of the doped films of the phenyl-substituted materials, which means that the triplet energy level can be easily utilized. This availability of the triplet level is expected to minimize the triplet-exciton loss of the host and increase the energy transfer to v-DABNA. As a result, the external heavy-atom effect of Si in the host material can increase the SOC and the kRISC of the v-DABNA dopant, which increases the EQE value of the doped device. The small roll-off characteristics of the devices under high luminance and high electric fields are also attributed to this fast energy transfer.

1-A. Comment: I do not understand this claim. Why does the host-to-dopant energy transfer process correlate with the delayed lifetime? The shorter delayed lifetime is simply attributed to higher kRISC of the dopant in the PL process, which is independent of the energy transfer in the EL process. The authors should discuss the PL and EL processes separately to clarify the point.

Answer: We thank the reviewer for these important comments; we agree with the reviewer's opinion. Ideally we need to explain the related PL and EL mechanisms separately because they differ. However, there have been several papers which explain EL performance data by the suppression of triplet-triplet or singlet-triplet annihilation based on fast kRISC in the PL (Nat. Photon. 2019, 13, 540 and Nat. Photon. 2019, 13, 678). Also, host materials including a relatively heavy atom can provide increased kRISC of the dopant, as shown by Prof. Baldo's group (Adv. Mater. 2017, 29, 1701987). According to the reviewer's comment, we amended and moved the related explanation to the part of the text describing the PL results. Instead of the explanation of EL performance in our original submission, in the revised manuscript we consider the energy transfer results and an explanation using Stern-Volmer experiment, even though the related data are derived from PL experiments, because this approach is frequently used when explaining EL performance.

	related text	cited page	paper#	corresponding author
1	the high EQE of the TDBA-DI emitter is attributed to its unique molecular structure such as (1) highly conjugated and rigid donor and acceptor combination for high PLQY and narrow emission, (2) large dihedral angle between the donor and acceptor for small ΔE_{ST} , fast kRISC rate and reduced self-quenching	#5, right column, line 14	Nat. Photon. 2019, 13, 540	Jang Hyuk Kwon
2	We assume that the relatively short TADF lifetime (4.05 μ s) and the large kRISC value ($2.0 \times 10^{-5} \text{ s}^{-1}$) suppress the triplet-triplet and singlet-triplet annihilation processes in the device.	#4, Left column, line 49	Nat. Photon. 2019, 13, 678	Takuji Hatakeyama
3	The improved roll-off behavior of the devices based on the brominated hosts is in agreement	#5, Right	Adv. Mater. 2017, 29,	Marc A. Baldo and Markus

	with the predicted alleviation of bimolecular loss processes like TTA, TCA, and singlet-triplet annihilation as a consequence of the lower steady state triplet concentrations.	column, line 3-13	1701987	Einzinger
4	The outstanding EQEmax of device D is attributed to a concomitant high Φ_{PL} (~100%) of EML and horizontal orientation factor ($\Theta_{ } = 89\%$; Supplementary Fig. 21a), and matches well with the theoretically predicted efficiency by using optical simulation	#3, right column, line 6	Nat. Photon. 2022, 16, 503	Chuluo Yang

Before)

The compounds with the TPS moiety showed an RISC speed 1.5 to 3.1 times faster than that of the compounds with a phenyl group, indicating that energy transfer occurs faster in the case of the TPS-substituted materials.

After)

=> The sentences were revised as below and added into PL part in Page 5, line 25

The compounds with the TPS moiety thus showed RISC speeds 1.5 to 3.1 times faster than those of the compounds with a phenyl group. The higher RISC speeds of the TPS-containing compounds might be explained by the heavy atom effect reported by Prof. Baldo et al³¹. Specifically, the external heavy-atom effect of Si in the host material can increase the k_{RISC} of the ν -DABNA dopant, leading to decreased triplet-exciton loss of the dopant and hence a higher EQE value of the doped device.

Before)

Possible explanations for the enhanced EL performance of the TPS-containing host materials compared with that of the phenyl-substituted host materials are as follows. First, the enhanced EL performance can be explained by the rapid energy transfer of the three TPS-substituted materials to the dopant. In the ν -DABNA-doped films of the three materials with the TPS moiety, the delayed lifetime was as much as three times shorter than that of the doped films of the phenyl-substituted materials, which means that the triplet energy level can be easily utilized. This availability of the triplet level is expected to minimize the triplet-exciton loss of the host and increase the energy transfer to ν -DABNA. As a result, the external heavy-atom effect of Si in the host material can increase the SOC and the k_{RISC} of the ν -DABNA dopant, which increases the EQE value of the doped device. The small roll-off characteristics of the devices under high luminance and high electric fields are also attributed to this fast energy transfer.

After)

=> It was removed.

=> Added the energy transfer result in terms of Stern-Volmer experiment in Page 7, line 16

Possible explanations for the enhanced EL performance of the TPS-based host materials compared with that of the TPS-free host materials are as follows. First, the enhanced EL performance can be explained by the rapid energy transfer from the TPS-based host materials to the dopant. As mentioned above in the discussion of PL, larger energy transfer values in the doped TPS-based films were confirmed by the results of Stern-Volmer experiments, and these larger energy transfer values could cause not only decreased triplet-exciton loss but also increased EL performance.

(1) By using the Stern-Volmer equation (energy transfer) and the related experiment, we added the related data as outlined below.

Added) Page 5, line 33

To confirm that efficient energy transfer occurs between the TPS-based host materials and ν -DABNA, we measured the energy transfer values of the Stern-Volmer equation (Supplementary Table S7). The k_q values of the three TPS-based host materials are larger than those of the corresponding TPS-free host materials.

Supplementary Table S7 | Rate constant of energy transfer between host and dopant based on Stern-Volmer equation.

	TDBA-Ph	mTDBA-Ph	mTDBA-2Ph	TDBA-Si	mTDBA-Si	mTDBA-2Si
Slope	0.13	0.048	0.14	0.43	0.19	0.16
k_q^a	3.43×10^7	1.02×10^7	3.23×10^7	9.48×10^7	4.08×10^7	5.90×10^7

^a Energy transfer that occurs between host and dopant: $I_0/I = 1 + k_{\text{quenching}} \times [A]$

1-B. Comment: Moreover, in Figure 3c, the efficiency roll-offs for three TPS-based host materials appear comparable to those for three TPS-free host materials. In particular, mTBDA-2Si, which exhibits the shortest delayed lifetime and highest kRISC value, exhibited the largest roll-off, which is inconsistent with the claim. The authors should discuss on the difference of EQE at lower current density rather than the roll-off at higher current density. The difference at lower current density may be simply explained by exciton formation efficiency. I assume that the TPS-free host materials have lower carrier mobilities than the three TPS-based host materials because of their smaller pai-skeleton with the bulky tert-butyl substituents. Therefore, it is strongly recommended to determine the carrier mobilities of the neat films of the host materials with/without the dopant.

Answer: We thank the reviewer for the insightful comments; we basically agree with the reviewer's opinion. Accordingly, we measured the related charge (hole and electron) mobility values and added the revised explanation as shown below. Also, as we mentioned in the 'Molecular design concept and synthesis' part of the first submission, TPS-based host materials have slower carrier mobilities than the corresponding TPS-free host materials because of the higher crystallinity of the latter materials, as shown in Fig S19, but the charge balance value can be used to explain the severe roll-off observed for the device with mTDBA-2Si as the host material.

Supplementary Fig. S19 | Optical microscopy image of the 2 wt% v-DABNA-doped TDDBA-based film. a TDDBA-Ph. b mTDDBA-Ph. c mTDDBA-2Ph. d TDDBA-Si. e mTDDBA-Si. f mTDDBA-2Si.

Added) Page 7, line 10

The more severe roll-off observed for mTDDBA-2Si compared to TDDBA-Si and mTDDBA-Si may be due to the higher imbalance in charge mobility of holes and electrons of mTDDBA-2Si (Supplementary Table S9). Specifically, the charge balance value, which is the hole mobility of the hole-only device (HOD) divided by the electron mobility of the electron-only device (EOD) in the doped devices, of TDDBA-Si, mTDDBA-Si, and mTDDBA-2Si was 1.05, 1.29, and 4.00, respectively. The imbalanced carrier mobility of holes and electrons in mTDDBA-2Si can cause severe EQE roll-off in device performance.

Supplementary Table S9 | Charge mobility of TDDBA-based host materials.

Compounds	Mobility (cm ² /Vs) @ 1V			
	Non-doped film		Doped film	
	HOD ^a	EOD ^b	HOD ^a	EOD ^b
TDDBA-Ph	1.59 x 10 ⁻⁵	1.01 x 10 ⁻⁶	8.71 x 10 ⁻⁷	7.79 x 10 ⁻⁸
mTDDBA-Ph	1.34 x 10 ⁻⁵	9.61 x 10 ⁻⁷	5.51 x 10 ⁻⁷	6.51 x 10 ⁻⁸
mTDDBA-2Ph	2.67 x 10 ⁻⁵	8.89 x 10 ⁻⁷	1.45 x 10 ⁻⁶	6.96 x 10 ⁻⁸
TDDBA-Si	5.02 x 10 ⁻⁷	7.04 x 10 ⁻⁷	3.18 x 10 ⁻⁸	3.04 x 10 ⁻⁸
mTDDBA-Si	2.83 x 10 ⁻⁷	7.03 x 10 ⁻⁷	2.39 x 10 ⁻⁸	1.85 x 10 ⁻⁸
mTDDBA-2Si	9.40 x 10 ⁻⁷	4.58 x 10 ⁻⁷	5.63 x 10 ⁻⁸	1.41 x 10 ⁻⁸

^aITO/MoO₃ (0.75 nm)/non-doped or doped (50 nm)/MoO₃ (1 nm)/Al (100 nm). ^bITO/TmPyPB (40 nm)/non-doped or doped (50 nm)/LIF (0.1 nm)/Al (100 nm).

2. Authors' claim: Second, the enhanced EL performance can be explained by the role of the hot triplet excited state. The existence of a T_n state similar to the S1 energy level might improve the EL performance through the activated RISC process.

2. Comment: This is possible, but not supported by the experimental data. As shown in Figure 1 and Table 1, the neat films of the TPS-free host materials showed shorter delayed lifetimes than those of the three TPS-based host materials. This indicates that the RISC process at the former is faster than those at the latter in the OLEDs. Moreover, the SOC and TD-DFT calculations in Figure S2–S8 do not clearly support this claim. Clear and careful arguments are required to convince the readers.

Answer: We agree with the reviewer's insightful comments. Regarding the delayed lifetimes of the doped TPS-based and TPS-free films, the former had a shorter delayed lifetime than the latter. Also, the paper of Prof. Baldo's group (Adv. Mater. 2017, 29, 1701987) showed that use of a heavy-atom host material in the doped film increased the film's k_{RISC} value. Moreover, in our original submission, we referred to the T_n level for the reversible intersystem crossing mechanism because there was a chance that both the T_3 and T_2 levels were involved. However, according to the reviewer's point, the delayed lifetimes of the TPS-free host materials were shorter than those of TPS-based materials. We will remove this claim from the current manuscript and separately report the related experimental data and discussion, including the dynamics of the excited state, in a future publication. In terms of the related experiments, such as transient absorption spectroscopy, we will require 2 months to 1 year to collect accurate data and analyze it. Based on the reviewer's suggestion that we compare the carrier mobilities of the materials, we added the related explanation to the second part of 'Possible explanations for the enhanced EL performance' subsection, as shown below.

Before)

Second, the enhanced EL performance can be explained by the role of the hot triplet excited state. The existence of a T_n state similar to the S1 energy level might improve the EL performance through the activated RISC process.

After text) Page 7, line 21

Second, the enhanced EL performance can be explained by balanced carrier mobility. When the carrier mobility values are compared between TPS-free host materials and TPS-based host materials, the latter shows the relatively superior property to the former, it causes the improved EL performance (Supplementary Table S9).

3. Authors' claim: Third, the enhanced EL performance of the TPS-containing host materials can be explained by their excellent thermal stability and surface properties (Supplementary Fig. S19-22). TDBA-Si, mTDBA-Si, and mTDBA-2Si exhibit a high T_g in the range 122–180 °C; they therefore do not easily undergo molecular changes because they maintain an amorphous state. The surface morphology of the deposited films will affect the device efficiency.

3. Comment: I do not agree with this claim. A T_g of 93 °C (mTBDA-2Ph) should be sufficient for the device fabrication and IVL measurement. Crystallization of the host material during the process significantly changes the EL properties at certain data points.

Answer: We thank the reviewer for this comment. We observed the time evolution of crystalline growth using optical microscopy (Reviewer only, Fig. R1). mTDBA-2Ph, with a T_g of 93 °C, showed a more stable film surface compared to TDBA-Ph and mTDBA-Ph, consistent with the reviewer's comments. However, the TPS-based host materials exhibited very stable amorphous film surfaces even at long storage times up to 24 h. We basically agree with the reviewer's comment because a T_g of 93 °C is sufficiently high to measure the EL performance immediately after device fabrication. Thus, we amended the text as shown below.

Before)

Third, the enhanced EL performance of the TPS-containing host materials can be explained by their excellent thermal stability and surface properties (Supplementary Fig. S19-22). TDBA-Si, mTDBA-Si, and mTDBA-2Si exhibit a high T_g in the range 122–180 °C; they therefore do not easily undergo molecular changes because they maintain an amorphous state. The surface morphology of the deposited films will affect the device efficiency

After) Page 7, line 27

Fourth, the enhanced EL performance of the TPS-containing host materials may be matched with their excellent thermal stability and surface properties (Supplementary Fig. S23-26). TDBA-Si, mTDBA-Si, and mTDBA-2Si exhibit high T_g values in the range of 122–180 °C, whereas mTDBA-2Ph, TDBA-Ph and mTDBA-Ph exhibit T_g values of 93, 62 and 58 °C, respectively. Thus, the T_g increment on going from TDBA-Ph and mTDBA-Ph to TDBA-Si and mTDBA-Si is 68 and 64 °C, respectively. As a result, the TPS-based host materials maintain an amorphous state well because they do not easily undergo molecular changes. The stable surface morphology of the deposited films maintains the device efficiency.

Fig. R1 Surface morphology over time in the film state using TDDBA-based host materials: (a) TDDBA-Ph, (b) mTDDBA-Ph, (c) mTDDBA-2Ph, (d) TDDBA-Si, (e) mTDDBA-Si, and (f) mTDDBA-2Si. (film thickness: 50 nm).

4. Authors' claim: Fourth, the enhanced EL performance can be explained by the high horizontal molecular dipole orientation. As previously mentioned, the horizontal orientation ratio of the three materials with the TPS moiety is very high (approximately 88–91%). As reported by Kwon's group, the highly horizontal orientation ratio is the basis for TDBA-Si exhibiting an EQE of 36% in simulations.

4. Comment: As shown in Figure 18, the horizontal orientation ratio for the TPS-based host materials (88-91%) is slightly higher than those (78-86%) of the TPS-free host materials. However, these small differences do not account for the experimental results.

Answer: We thank the reviewer for this comment. Accordingly, we amended the text related to the detailed horizontal orientation increment values. We list the related previous reference papers below.

	paper	corresponding author	cited page
1	Adv. Sci., 2022, 9, 2203903	Jun Yeob Lee	#7, left column, line 12
2	Adv. Funct. Mater., 2013, 23, 3896–3900	Jang-Joo Kim	#4, right column, line 9
3	Nat. Photonics., 2019, 13, 540-546	Jang Hyuk Kwon	#5, left column, line 13

Before)

Fourth, the enhanced EL performance can be explained by the high horizontal molecular dipole orientation. As previously mentioned, the horizontal orientation ratio of the three materials with the TPS moiety is very high (approximately 88–91%). As reported by Kwon's group, the highly horizontal orientation ratio is the basis for TDBA-Si exhibiting an EQE of 36% in simulations.

After) Page 7, line 222

Third, the enhanced EL performance can be explained by the high horizontal molecular dipole orientation. As previously mentioned, the horizontal orientation ratios of TDBA-Ph, mTDBA-Ph, and mTDBA-2Ph were 84%, 78%, and 86%, while TDBA-Si, mTDBA-Si, and mTDBA-2Si were 91%, 88%, and 89%, indicating an increment of 7%, 10%, and 3% respectively. As reported by Prof. Kwon's group, the high horizontal orientation ratio is the basis for TDBA-Si exhibiting an EQE of 36% in simulations¹⁰.

Before)

It may also be explained by a T₂ excited-state contribution, high thermal stability, and high horizontal orientation.

After) Page 1, line 30

Other factors possibly contributing to the high performance are a T₂ excited-state contribution, high horizontal orientation, and high thermal stability.

REVIEWER COMMENTS

Reviewer #1 (Remarks to the Author):

Park et al. have extensively revised the manuscript, particularly on the photophysical mechanism of TADF host-TADF guest system. As already mentioned in the first revision, the state-of-the-art device performance in combination with a series of silicon-based hosts provide a new host choice for the OLED community, and hence, further revisions are not required after this round of revision. However, some of the photophysical interpretations at current format may not meet the fundamental principle or experimental observation. Therefore, the authors are recommended to revise the text accordingly, and if possible, to remind the audiences an operation of an alternative TADF mechanism.

First of all, in terms of RISC from highly excited state, the authors seemed to have misunderstood our previous suggestions. Although the authors have replaced T_n with T₂ throughout the text, T₂ still belongs to the highly excited state, namely, T_n. One of the most important criteria for T₂ to participate in the RISC process is the nearly degenerate (via vibronic coupling) T₂ and T₁ energy levels, where the T₂-T₁ energy gap generally less than 0.05 eV (ChemPhysChem 2016, 17, 2956–2961). An elegant experimental evidence could also be found in Nat Commun 7, 13680 (2016), where the report demonstrated a manipulation on the 3CT relative to the 3LE state energy with polyethylene oxide. As for the titled host materials, it lacks the fundamental support for the RISC process to occur from T₂ to S₁ (Supplementary Fig. S3 ~ S9). Additionally, their experimental ΔE_{S1T1} s are clearly seen in Supplementary Fig. S19 (~0.27 eV), thus the calculated ΔE_{S1T1} s (> 0.3 eV) are plausibly overestimated due to the uncertainty of calculated triplet energy. Furthermore, even if the T₁ population somehow populated to T₂, with a T₂-T₁ energy gap less than 1 eV, the ultrafast internal conversion would dominate the kinetics of T₂ and leave RISC no chance (or negligibly) to take place. Therefore, please correct every statement regarding “RISC from T₂ to S₁” into “RISC from T₁ to S₁”.

Next, in terms of the excited state mechanism, there is no doubt that the Förster type energy transfer is operative in this study (Supplementary Fig. S21). However, after careful reevaluating the triexponential kinetics in Figure 2b along with the provided FRET rate constants (~10 ns), I am now convinced that the second (τ_2) and third fitted decay (τ_3) may stem from the TADF kinetics of either the host or dopant. Two limiting cases could be interpreted as follow. If the delayed fluorescence of host is much slower than the that of the dopant, under the current kinetic scheme, τ_3 and τ_2 indicate the host's and dopant's triplet state lifetime, respectively. On the contrary, if the delayed fluorescence of dopant is slower, then the meaning of τ_3 and τ_2 would switch. Nonetheless, the fitted τ_2 and τ_3 in Supplementary Table S5 fail to correspond to the photophysics of independent hosts and dopants. In combination with the absence of ground state complex in the doped film, the operation of host-guest interaction in the excited state becomes highly possible. If the authors are not able to exclude this possibility with transient absorption at current stage, I recommend the authors at least cite this reference (Nature

Photonics 15.10 (2021): 780-786) and state the possible exciplex formation in the main text in order to explain the anomalous rate constants in Figure 2b and Supplementary Table 5.

Reviewer #2 (Remarks to the Author):

The author teams have nicely and comprehensively addressed the comments to satisfaction and I have no other concern. It can be accepted for publication.

Reviewer #3 (Remarks to the Author):

The manuscript was improved, but I do not recommend publication.

(See attachment)

According to this reviewer's original comments, the authors have measured HOD/EOD devices (Table S9) to confirm that the charge balance of TPS-based hosts is better than that of Ph-based hosts, nicely accounting for the improved maximum EQE. However, the authors keep the following claim.

The higher RISC speeds of the TPS-containing compounds might be explained by the heavy atom effect reported by Prof. Baldo et al. Specifically, the external heavy-atom effect of Si in the host material can increase the k_{RISC} of the ν -DABNA dopant, leading to decreased triplet-exciton loss of the dopant and hence a higher EQE value of the doped device. The SOC was improved by the heavy-atom effect of Si atoms, similar to the improvement observed as a result of the heavy-atom effect of S and Se atoms in MR-TADF.

The heavy atom effect on the RISC is not so significant but supported by the experimental data. However, the effect on the EQE is suspicious. As shown in Table 3, the EQE for Ph-based TDBAs (12.9% to 10.3%, 13.0% to 10.4%, 8.2% to 7.2%) are comparable to TPS-based TDBAs (36.2% to 31.3%, 27.3% to 24.1%, 38.1% to 16.6%). In addition, mTDBA-2Si having the highest k_{RISC} ($8.64 \times 10^5 \text{ s}^{-1}$) shows the largest roll-off (38.1% to 16.6%). These results indicate that the TPS group improves the maximum EQE but not the roll-off. Notably, the original report for ν -DABNA-based OLED used DOBNA-OAr (Table S10, Nat. Photonics 2019) shows high EQE of 34.4% (to 26.0%) comparable the OLEDs with TPS-TDBAs. Since DOBNA-OAr has the same core structure as TDBA (TDBA is tert-Bu-substituted DOBNA), but no heavy atom, the contribution of the heavy atom effect on EQE should be declined.

Supplementary Table S9 | Charge mobility of TDBA-based host materials.

Compounds	Mobility (cm^2/Vs) @ 1V			
	Non-doped film		Doped film	
	HOD ^a	EOD ^b	HOD ^a	EOD ^b
TDBA-Ph	1.59×10^{-5}	1.01×10^{-6}	8.71×10^{-7}	7.79×10^{-8}
mTDBA-Ph	1.34×10^{-5}	9.61×10^{-7}	5.51×10^{-7}	6.51×10^{-8}
mTDBA-2Ph	2.67×10^{-5}	8.89×10^{-7}	1.45×10^{-6}	6.96×10^{-8}
TDBA-Si	5.02×10^{-7}	7.04×10^{-7}	3.18×10^{-8}	3.04×10^{-8}
mTDBA-Si	2.83×10^{-7}	7.03×10^{-7}	2.39×10^{-8}	1.85×10^{-8}
mTDBA-2Si	9.40×10^{-7}	4.58×10^{-7}	5.63×10^{-8}	1.41×10^{-8}

Table 2 | Photophysical, thermal, and surface properties of doped films.

2 wt% ν -DABNA-doped film					
	$\theta_{//}^a$ (%)	PLQY	τ_d^b (μ s)	$\lambda_{max}/FWHM$ (nm)	k_{RISC} (10^5 s $^{-1}$)
TDBA-Ph	84	0.82	2.99	469/18	2.25
mTDBA-Ph	78	0.93	2.76	469/18	3.13
mTDBA-2Ph	86	0.86	1.86	468/18	3.98
TDBA-Si	91	0.82	0.96	469/18	7.01
mTDBA-Si	88	0.94	1.93	469/18	4.58
mTDBA-2Si	89	0.92	0.98	468/18	8.64

^a Horizontal transient dipole orientation ratio. ^b Delayed lifetime calculated by photoluminescence decay. ^c Glass transition temperature. ^d Decomposition temperature. ^e Root mean square roughness measured by atomic force microscopy (AFM).

Table 3 | EL performance of 2 wt% ν -DABNA-doped OLED devices according to the host materials.

EMLs	T_{on}^a (V)	L_{max} (cd m $^{-2}$)	CE ^b (cd A $^{-1}$)			EQE ^c (%)		
			Max.	500 cd m $^{-2}$	1,000 cd m $^{-2}$	Max.	500 cd m $^{-2}$	1,000 cd m $^{-2}$
TDBA-Ph: ν -DABNA	3.10	4,400	10.8	9.80	8.48	12.9	11.8	10.3
mTDBA-Ph: ν -DABNA	3.09	4,500	10.9	9.81	8.48	13.0	11.8	10.4
mTDBA- 2Ph: ν -DABNA	3.06	5,900	7.24	6.82	6.34	8.21	7.75	7.24
TDBA-Si: ν -DABNA	3.00	7,300	38.1	36.2	31.8	36.2	35.0	31.3
mTDBA-Si: ν -DABNA	3.02	7,100	27.9	27.5	24.1	27.3	26.9	24.1
mTDBA-2Si: ν -DABNA	3.00	6,300	50.3	37.5	21.5	38.1	28.8	16.6

^a Turn-on voltage at 1 cd m⁻². ^b Current efficiency. ^c External quantum efficiency. ^d Commission Internationale de l'Eclairage.

I do not understand the claim “In a two-component system, the newly synthesized host materials may facilitate the FRET process to the dopant through RISC activation of host T₂ level.”. If the authors suppose TADF-OLEDs, triplet excitons should be mostly up-converted in the emitter (v-DABNA), not in the host materials. Therefore, FRET from the host to the TADF could not be the major reason for the drastic improvement of EQE. If the authors suppose TAF-OLEDs (so-called hyperfluorescence system) where the host material acts as an assisting dopant, the faster FRET from the host to the TADF can be the main reason for the improvement. However, I could not find any correlation between kRISC (Table S4) of the host materials and EQE (Table 3). The authors should clarify the EL process in the device and clearly discuss the role of the host material. Moreover, the SOCs in Figure S4-S9 and the ST-gap in Figure S19 do not correlate with the kRISCs in Table S4. Especially, for TBDA-Ph (SOC of 0.01 cm⁻¹ and the ST-gap of ~0.20 eV), kRISC could not be 4.23 x 10⁵ s⁻¹. The authors should provide TRPL of non-doped films of TBDAs to prove the experimental data. There would be mistakes in the TRPL fitting or related calculations.

Supplementary Table S4 | Rate constant for TDBA based host materials (non-doped film) at room temperature.

	TDBA-Ph	mTDBA-Ph	mTDBA-2Ph	TDBA-Si	mTDBA-Si	mTDBA-2Si
Φ	0.62	0.65	0.63	0.66	0.61	0.75
Φ_F	0.538	0.589	0.603	0.644	0.592	0.731
Φ_{TADF}	0.082	0.061	0.027	0.016	0.018	0.019
τ_f (ns)	9.35	10.01	8.92	7.80	11.1	6.50
τ_{TADF} (μ s)	0.91	1.25	2.33	3.53	6.65	6.49
k_F ($\times 10^7$)	5.75	5.87	6.76	8.25	5.32	11.2
k_{IC} ($\times 10^7$)	3.53	3.16	3.97	4.25	3.40	3.75
k_{ISC} ($\times 10^7$)	10.4	9.27	4.85	3.14	2.63	3.92
Φ_{IC}	0.33	0.317	0.354	0.332	0.379	0.244
Φ_{ISC}	0.13	0.09	0.04	0.02	0.03	0.03
k_{TADF} ($\times 10^5$)	6.82	5.22	2.71	1.87	9.18	1.16
k_{RISC} ($\times 10^5$)	4.23	3.39	1.71	1.23	5.60	8.66

Supplementary Fig. S19 | Absorption spectra(dash) at room temperature, low temperature photoluminescence without (line) and with delay (dot) at 77K spectra of neat film. a TDBA-Ph. b mTDBA-Ph. c mTDBA-2Ph. d TDBA-Si. e mTDBA-Si. f mTDBA-2Si.

Supplementary Fig. S4 | Isosurface of HOMO and LUMO composing $S_0 \rightarrow S_1$ transition (isovalue = 0.02) with representative electronic transition energies with SOC values of TDBA-Ph. TD-B3LYP calculation was conducted at the level of 6-31G(d,p).

Supplementary Fig. S5 | Isosurface of HOMO and LUMO composing $S_0 \rightarrow S_1$ transition (isovalue = 0.02) with representative electronic transition energies with SOC values of mTDBA-Ph. TD-B3LYP calculation was conducted at the level of 6-31G(d,p).

Supplementary Fig. S6 | Isosurface of HOMO and LUMO composing $S_0 \rightarrow S_1$ transition (isovalue = 0.02) with representative electronic transition energies with SOC values of mTDBA-2Ph. TD-B3LYP calculation was conducted at the level of 6-31G(d,p).

Supplementary Fig. S7 | Isosurface of HOMO and LUMO composing $S_0 \rightarrow S_1$ transition (isovalue = 0.02) with representative electronic transition energies with SOC values of TDBA-Si. TD-B3LYP calculation was conducted at the level of 6-31G(d,p).

Supplementary Fig. S8 | Isosurface of HOMO and LUMO composing $S_0 \rightarrow S_1$ transition (isovalue = 0.02) with representative electronic transition energies with SOC values of mTDBA-Si. TD-B3LYP calculation was conducted at the level of 6-31G(d,p).

Supplementary Fig. S9 | Isosurface of HOMO and LUMO composing $S_0 \rightarrow S_1$ transition (isovalue = 0.02) with representative electronic transition energies with SOC values of mTDBA-Si. TD-B3LYP calculation was conducted at the level of 6-31G(d,p).

REVIEWER COMMENTS

Reviewer #1 (Remarks to the Author):

Park et al. have extensively revised the manuscript, particularly on the photophysical mechanism of TADF host-TADF guest system. As already mentioned in the first revision, the state-of-the-art device performance in combination with a series of silicon-based hosts provide a new host choice for the OLED community, and hence, further revisions are not required after this round of revision. However, some of the photophysical interpretations at current format may not meet the fundamental principle or experimental observation. Therefore, the authors are recommended to revise the text accordingly, and if possible, to remind the audiences an operation of an alternative TADF mechanism.

First of all, in terms of RISC from highly excited state, the authors seemed to have misunderstood our previous suggestions. Although the authors have replaced T_n with T₂ throughout the text, T₂ still belongs to the highly excited state, namely, T_n. One of the most important criteria for T₂ to participate in the RISC process is the nearly degenerate (via vibronic coupling) T₂ and T₁ energy levels, where the T₂-T₁ energy gap generally less than 0.05 eV (ChemPhysChem 2016, 17, 2956–2961). An elegant experimental evidence could also be found in Nat Commun 7, 13680 (2016), where the report demonstrated a manipulation on the 3CT relative to the 3LE state energy with polyethylene oxide. As for the titled host materials, it lacks the fundamental support for the RISC process to occur from T₂ to S₁ (Supplementary Fig. S3 ~ S9). Additionally, their experimental $\Delta E_{S_1T_1}$ s are clearly seen in Supplementary Fig. S19 (~0.27 eV), thus the calculated $\Delta E_{S_1T_1}$ s (> 0.3 eV) are plausibly overestimated due to the uncertainty of calculated triplet energy. Furthermore, even if the T₁ population somehow populated to T₂, with a T₂-T₁ energy gap less than 1 eV, the ultrafast internal conversion would dominate the kinetics of T₂ and leave RISC no chance (or negligibly) to take place. Therefore, please correct every statement regarding “RISC from T₂ to S₁” into “RISC from T₁ to S₁”.

Answer: We thank the reviewer for these insightful comments. We agree with the reviewer's opinion. We modified all references to the RISC process from T₂ to S₁ to the RISC process from T₁ to S₁ in the text, and the figures related to the text have been modified. In addition, several important references were added.

Added text) Page 3, line 115

However, Monkman group reported a well-defined photodynamic mechanism for the RISC process at the T₁ and T₂ levels^{24,25}. On the basis of these findings, the direct RISC process from the T₂ level to the S₁ level will not occur in the system examined in the present study, and the transition from the T₁ level to the S₁ level will be the main process.

Ref 24 : Nature Communications., 7, 13680 (2016). Revealing the spin–vibronic coupling mechanism of thermally activated delayed fluorescence

Ref 25 : ChemPhysChem., 17, 2956–2961 (2016). The Importance of Vibronic Coupling for Efficient Reverse Intersystem Crossing in Thermally Activated Delayed Fluorescence Molecules

Revised text)

No.	Position	Before	After
1	Page 1, line 28	This high performance is attributed to fast energy transfer from the host to the dopant, which is enabled by the external heavy-atom effect of Si, inhibition of aggregation by the bulky tetraphenylsilyl groups, and fast reverse intersystem crossing of the dopant. Other factors possibly contributing to the high performance are a T ₂ excited-state contribution, high horizontal orientation, and high thermal stability.	This high performance is attributed to fast energy transfer from the host to the dopant. Other factors possibly contributing to the high performance are a T ₁ excited-state contribution, inhibition of aggregation by the bulky tetraphenylsilyl groups, high horizontal orientation, and high thermal stability.
2	Page 2, line 74	The spin-orbit coupling (SOC) of the S ₁ and T ₁ states and that of the S ₁ and T ₂ states for the materials were calculated and compared. Energy transfer from the new host materials to the dopant was confirmed to occur via the Förster resonance energy transfer (FRET) process through the T ₂ state to the singlet states.	The spin-orbit coupling (SOC) of the S ₁ and T ₁ states and of the S ₁ and T ₂ states of the materials were calculated. Energy transfer from the new host materials to the dopant was confirmed to occur via Förster resonance energy transfer (FRET) through the T ₁ state to the singlet states.
3	Page 4, line 137	Although the ΔE_{ST} of S ₁ -T ₁ is relatively large (e.g., >0.2 eV), the T ₂ state and the S ₁ energy gap are relatively small, which can be interpreted as promoting RISC from the T ₂ state to the S ₁ level. This result means that the new host materials may promote activation of the RISC process because of the combination of the contribution of the T ₂ state and the TADF properties.	Although ΔE_{ST} is smaller for S ₁ -T ₂ than for S ₁ -T ₁ (Supplementary Table S2), the RISC process occurs only in the T ₁ state because of rapid internal conversion from T ₂ to T ₁ .
4	Page 5, line 178	In a two-component system, the newly synthesized host materials may facilitate the FRET process to the dopant through RISC activation of host T ₂ level	In a two-component system, the newly synthesized host materials may facilitate the FRET process to the dopant through RISC activation of the host T ₁ level.
5	Page 3, line 115 (in 2 nd revision) / Page 4, line 118 (in 1 st revision file)	On the basis of the reports of Hatakeyama's group, the newly synthesized host materials are expected to exhibit TADF properties and to utilize the T ₂ energy level ^{8,26-28} . In addition, most of the materials have larger SOC values of $\langle T_2 \rangle$ between the S ₁ and T ₂ levels compared with the SOC value of $\langle T_1 \rangle$ between the S ₁ and T ₁ levels (Supplementary Fig. S3-S9 and Table S3). Recently several important studies have examined systems with small SOC values (< 1 cm ⁻¹) ^{5,28,29} . The six materials examined in the present work showed higher SOC values than in these previous works. These results show that the newly synthesized TDBA-based host materials have the advantage of increased luminous efficiency through	These sentences have been deleted.

	activation of RISC by simultaneously utilizing the T ₂ state as well as conventional TADF characteristics.	
--	---	--

Revised Figures)

Supplementary Fig. S3 | Isosurface of HOMO and LUMO composing S₀→S₁ transition (isovalue = 0.02) with representative electronic transition energies with SOC values of TDBA. TD-B3LYP calculation was conducted at the level of 6-31G(d,p).

Supplementary Fig. S4 | Isosurface of HOMO and LUMO composing S₀→S₁ transition (isovalue = 0.02) with representative electronic transition energies with SOC values of TDBA-Ph. TD-B3LYP calculation was conducted at the level of 6-31G(d,p).

Supplementary Fig. S5 | Isosurface of HOMO and LUMO composing $S_0 \rightarrow S_1$ transition (isovalue = 0.02) with representative electronic transition energies with SOC values of mTDBA-Ph. TD-B3LYP calculation was conducted at the level of 6-31G(d,p).

Supplementary Fig. S6 | Isosurface of HOMO and LUMO composing $S_0 \rightarrow S_1$ transition (isovalue = 0.02) with representative electronic transition energies with SOC values of mTDBA-2Ph. TD-B3LYP calculation was conducted at the level of 6-31G(d,p).

Supplementary Fig. S7 | Isosurface of HOMO and LUMO composing $S_0 \rightarrow S_1$ transition (isovalue = 0.02) with representative electronic transition energies with SOC values of TDDBA-Si. TD-B3LYP calculation was conducted at the level of 6-31G(d,p).

Supplementary Fig. S8 | Isosurface of HOMO and LUMO composing $S_0 \rightarrow S_1$ transition (isovalue = 0.02) with representative electronic transition energies with SOC values of mTDDBA-Si. TD-B3LYP calculation was conducted at the level of 6-31G(d,p).

Supplementary Fig. S9 | Isosurface of HOMO and LUMO composing $S_0 \rightarrow S_1$ transition (isovalue = 0.02) with representative electronic transition energies with SOC values of mTDBA-Si. TD-B3LYP calculation was conducted at the level of 6-31G(d,p).

Fig. 3 | **a** Schematic of the conceived energy-transfer mechanism for the host and the ν -DABNA dopant. **b** Energy-level diagram of doped OLED devices. **c** Molecular structures used in each layer. **d** Current efficiency (CE)–luminance (L) curves. **e** External quantum efficiency (EQE)–L curves. **f** EL spectra (inset: OLED driving image at 7 V).

Next, in terms of the excited state mechanism, there is no doubt that the Förster type energy transfer is operative in this study (Supplementary Fig. S21). However, after careful reevaluating the triexponential kinetics in Figure 2b along with the provided FRET rate constants (~ 10 ns), I am now convinced that the second (τ_2) and third fitted decay (τ_3) may stem from the TADF kinetics of either the host or dopant. Two limiting cases could be interpreted as follow. If the delayed fluorescence of host is much slower than that of the dopant, under the current kinetic scheme, τ_3 and τ_2 indicate the host's and dopant's triplet state lifetime, respectively. On the contrary, if the delayed fluorescence of dopant is slower, then the meaning of τ_3 and τ_2 would switch.

Answer: We thank the reviewer for these insightful comments. We agree with the reviewer's opinion and rewrote the related text as follows:

Before)

As a result of the transient PL measurement in the doped films state, the exciton decay behavior can be divided into three exponential decay components (Supplementary Table S5). The energy transfer rate from host material to dopant corresponds to the second lifetime (τ_2) and the population decay of the dopant's triplet state corresponds to the third lifetime (τ_3). Comparing the second exponential decay values (τ_2) of the TPS-based hosts with TPS-free hosts, TDBA-Si and mTDBA-Si showed slightly faster decay than TDBA-Ph and mTDBA-Ph, respectively.

After) Page 5, line 160

Also, based on the results of the transient PL measurements in the doped film state, the exciton decay behavior can be divided into three exponential decay components (Supplementary Table S5). The second (τ_2) and third (τ_3) fitted decay times can be interpreted as corresponding to the dopant and host components, respectively. This interpretation is because not only the FRET rate constants between the synthesized host materials and ν -DABNA are in the range of $1.07 \sim 2.08 \times 10^8$ /s, as shown in Supplementary Table S8, but also the delayed fluorescence of the host is slower than that of the dopant.

Nonetheless, the fitted τ_2 and τ_3 in Supplementary Table S5 fail to correspond to the photophysics of independent hosts and dopants. In combination with the absence of ground state complex in the doped film, the operation of host-guest interaction in the excited state becomes highly possible. If the authors are not able to exclude this possibility with transient absorption at current stage, I recommend the authors at least cite this reference (Nature Photonics 15.10 (2021): 780-786) and state the possible exciplex formation in the main text in order to explain the anomalous rate constants in Figure 2b and Supplementary Table 5.

Answer: We thank the reviewer for these insightful comments. We agree with the reviewer's opinion and added the following text.

Added text) Page 5, line 166

The anomalous rate constants in Fig. 2b and Supplementary Table S5 can be attributed to exciplex formation of the host-dopant system. The ground state complex was not observed in the doped film, which made host-guest interaction in the excited state highly possible. As reported by Chou group, transient absorption experiments need to be conducted to examine whether exciplexes form; we will report the excitation dynamics data in a forthcoming paper²⁷.

Ref 27 : Nature Photonics., 15, 780-786 (2021)

Reviewer #2 (Remarks to the Author):

The author teams have nicely and comprehensively addressed the comments to satisfaction and I have no other concern. It can be accepted for publication.

Answer) Thank you for your insight and consideration.

Reviewer #3 (Remarks to the Author):

According to this reviewer's original comments, the authors have measured HOD/EOD devices (Table S9) to confirm that the charge balance of TPS-based hosts is better than that of Ph-based hosts, nicely accounting for the improved maximum EQE. However, the authors keep the following claim. The higher RISC speeds of the TPS-containing compounds might be explained by the heavy atom effect reported by Prof. Baldo et al. Specifically, the external heavy-atom effect of Si in the host material can increase the k_{RISC} of the *n*-DABNA dopant, leading to decreased triplet-exciton loss of the dopant and hence a higher EQE value of the doped device. The SOC was improved by the heavy-atom effect of Si atoms, similar to the improvement observed as a result of the heavy-atom effect of S and Se atoms in MR-TADF. The heavy atom effect on the RISC is not so significant but supported by the experimental data. However, the effect on the EQE is suspicious. As shown in Table 3, the EQE for Ph-based TBDAs (12.9% to 10.3%, 13.0% to 10.4%, 8.2% to 7.2%) are comparable to TPS-based TBDAs (36.2% to 31.3%, 27.3% to 24.1%, 38.1% to 16.6%). In addition, mTDBA-2Si having the highest k_{RISC} ($8.64 \times 10^5 \text{ s}^{-1}$) shows the largest roll-off (38.1% to 16.6%). These results indicate that the TPS group improves the maximum EQE but not the roll-off. Notably, the original report for *v*-DABNA-based OLED used DOBNA-OAr (Table S10, Nat. Photonics 2019) shows high EQE of 34.4% (to 26.0%) comparable the OLEDs with TPS-TBDAs. Since DOBNA-OAr has the same core structure as TDBA (TDBA is tert-Bu-substituted DOBNA), but no heavy atom, the contribution of the heavy atom effect on EQE should be declined.

Answer: We thank the reviewer for these insightful comments. We agree with the reviewer's opinion. Below we separately address the related issues.

Issue 1: Heavy atom effect of the Si atom

We agree that, in contrast to what was reported previously for S or Se, the heavy atom effect of Si does not contribute to the enhancement of SOC. Therefore, the text regarding the heavy atom effect and the related description were deleted in the manuscript.

Revised text)

No.	Position	Before	After
1	Page 1, line 28	This high performance is attributed to fast energy transfer from the host to the dopant, which is enabled by the external heavy-atom effect of Si, inhibition of aggregation by the bulky tetraphenylsilyl groups, and fast reverse intersystem crossing of the dopant.	This high performance is attributed to fast energy transfer from the host to the dopant. Other factors possibly contributing to the high performance are a T_1 excited-state contribution, inhibition of aggregation by the bulky tetraphenylsilyl groups, high horizontal orientation, and high thermal stability.
2	Page 2, line 71	These atoms have two functions: they bring into play the external heavy atom effect and they increase the intermolecular distance in the host material.	These atoms have the function of increasing the intermolecular distance in the host material.
3	Page 2, line 76 (in 2 nd revision) / Page 2, line 76 (in 1 st revision)	In particular, when TDBA-Si, mTDBA-Si, and mTDBA-2Si are doped with v -DABNA, the SOC of the dopant can be improved because of the external heavy-atom effect of the Si atom of the host and the rate constant for RISC (k_{RISC}) can become high ^{17,18} .	This sentence has been deleted.
4	Page 5,	Specifically, the external heavy-atom effect of Si in the host material can increase the	These sentences have been deleted.

	line 178 (in 2 nd revision) / Page 5, line 182 (in 1 st revision)	k_{RISC} of the ν -DABNA dopant, leading to decreased triplet-exciton loss of the dopant and hence a higher EQE value of the doped device. The SOC was improved by the heavy-atom effect of Si atoms, similar to the improvement observed as a result of the heavy-atom effect of S and Se atoms in MR-TADF ^{17,18,32-34} .	
5	Page 5, line 178 (in 2 nd revision) / Page 5, line 181 (in 1 st revision)	The higher RISC speeds of the TPS-containing compounds might be explained by the heavy atom effect reported by Prof. Baldo et al ³¹ .	This sentence has been deleted.
6	Page 3, line 115 (in 2 nd revision) / Page 4, line 118 (in 1 st revision)	On the basis of the reports of Hatakeyama's group, the newly synthesized host materials are expected to exhibit TADF properties and to utilize the T ₂ energy level ^{8,26-28} . In addition, most of the materials have larger SOC values of $\langle T_2 \rangle$ between the S ₁ and T ₂ levels compared with the SOC value of $\langle T_1 \rangle$ between the S ₁ and T ₁ levels (Supplementary Fig. S3–S9 and Table S3). Recently several important studies have examined systems with small SOC values (< 1 cm ⁻¹) ^{5,28,29} . The six materials examined in the present work showed higher SOC values than in these previous works. These results show that the newly synthesized TDBA-based host materials have the advantage of increased luminous efficiency through activation of RISC by simultaneously utilizing the T ₂ state as well as conventional TADF characteristics.	These sentences have been deleted.
7	Page 7, line 264	In addition, upon introduction of the TPS moiety, effective energy transfer to the dopant was confirmed by the external heavy-atom effect.	In addition, upon introduction of the TPS moiety, effective energy transfer to the dopant was observed.
8	Page 7, line 269	The molecular design concept in which a heavy atom introducing TPS is substituted to obtain an MR-TADF-type derivative is expected to facilitate the development of TADF hosts for high-performance blue OLEDs.	The molecular design concept in which TPS is introduced to obtain an MR-TADF-type derivative is expected to facilitate the development of TADF hosts for high-performance blue OLEDs.

Issue 2: Severe roll-off of mTDBA-2Si

In the first revision stage, the explanation in terms of roll-off of mTDBA-2Si was as follows:

“The more severe roll-off observed for mTDBA-2Si compared to TDBA-Si and mTDBA-Si may be due to the higher imbalance in charge mobility of holes and electrons of mTDBA-2Si (Supplementary Table S9). Specifically, the charge balance value, which is the hole mobility of the hole-only device (HOD) divided by the electron mobility of the electron-only device (EOD) in the doped devices, of TDBA-Si, mTDBA-Si, and mTDBA-2Si was 1.05, 1.29, and 4.00, respectively. The imbalanced carrier mobility of holes and electrons in mTDBA-2Si can cause severe EQE roll-off in device performance.”

We judge this to be correct, and hence have left it unchanged in the second revision stage. If you have

a different opinion, please let us know and we will seek to revise the text to address the issue.

Issue 3: High EQE and roll-off issue of the DOBNA-OAr host material (Reviewer only)

As mentioned by the reviewer, DOBNA-OAr, a host used in a previously reported paper on v-DABNA, was synthesized and its morphology after deposition was compared with that of TDBA-Ph, mTDBA-Ph, mTDBA-2Ph, TDBA-Si, mTDBA-Si, and mTDBA-2Si. As shown in the Fig. S23 and R1 below, in our work the TPS-free host materials formed crystals, but the neat films of DOBNA-OAr and TPS-based host materials did not crystallize and showed a smooth surface. Based on these findings, it can be concluded that DOBNA-OAr has a bulky structure that inhibits crystallization and that DOBNA-OAr films have an excellent surface morphology. As a result, devices based on DOBNA-OAr do not exhibit severe roll-off. If you need further corrections of this, please let us know.

Supplementary Fig. S23 | Optical microscopy image of the 2 wt% v-DABNA-doped TDBA-based film. a TDBA-Ph. b mTDBA-Ph. c mTDBA-2Ph. d TDBA-Si. e mTDBA-Si. f mTDBA-2Si.

Reviewer only Fig. R1 | Optical microscopy images of the non-doped DOBNA-OAr film at different times.

I do not understand the claim

“In a two-component system, the newly synthesized host materials may facilitate the FRET process to the dopant through RISC activation of host T2 level.”. If the authors suppose TADF-OLEDs, triplet excitons should be mostly up-converted in the emitter (*v*-DABNA), not in the host materials. Therefore, FRET from the host to the TADF could not be the major reason for the drastic improvement of EQE. If the authors suppose TADF-OLEDs (so-called hyperfluorescence system) where the host material acts as an assisting dopant, the faster FRET from the host to the TADF can be the main reason for the improvement. However, I could not find any correlation between *k*RISC (Table S4) of the host materials and EQE (Table 3). The authors should clarify the EL process in the device and clearly discuss the role of the host material. Moreover, the SOCs in Figure S4-S9 and the ST-gap in Figure S19 do not correlate with the *k*RISCs in Table S4. Especially, for TBDA-Ph (SOC of 0.01 cm⁻¹ and the ST-gap of ~0.20 eV), *k*RISC could not be 4.23 x 10⁵ s⁻¹. The authors should provide TRPL of non-doped films of TBDAs to prove the experimental data. There would be mistakes in the TRPL fitting or related calculations.

Answer: We acknowledge and agree with the reviewer's comments. The related sentences have been corrected and an interpretation of EQE has been added as follows.

Issue 1: RISC process modification of 'from T1 to S1' transition

In agreement with the reviewer's opinion, we corrected the related text.

Added text) Page 3, line 115

However, Monkman group reported a well-defined photodynamic mechanism for the RISC process at the T_1 and T_2 levels^{24,25}. On the basis of these findings, the direct RISC process from the T_2 level to the S_1 level will not occur in the system examined in the present study, and the transition from the T_1 level to the S_1 level will be the main process.

Ref 24 : Nature Communications., 7, 13680 (2016). Revealing the spin–vibronic coupling mechanism of thermally activated delayed fluorescence

Ref 25 : ChemPhysChem., 17, 2956 –2961 (2016). The Importance of Vibronic Coupling for Efficient Reverse Intersystem Crossing in Thermally Activated Delayed Fluorescence Molecules

Revised text)

No.	Position	Before	After
1	Page 1, line 28	This high performance is attributed to fast energy transfer from the host to the dopant, which is enabled by the external heavy-atom effect of Si, inhibition of aggregation by the bulky tetraphenylsilyl groups, and fast reverse intersystem crossing of the dopant. Other factors possibly contributing to the high performance are a T_2 excited-state contribution, high horizontal orientation, and high thermal stability.	This high performance is attributed to fast energy transfer from the host to the dopant. Other factors possibly contributing to the high performance are a T_1 excited-state contribution, inhibition of aggregation by the bulky tetraphenylsilyl groups, high horizontal orientation, and high thermal stability.
2	Page 2, line 74	The spin-orbit coupling (SOC) of the S_1 and T_1 states and that of the S_1 and T_2 states for the materials were calculated and compared. Energy transfer from the new host materials to the dopant was confirmed to occur via the Förster resonance energy transfer (FRET) process through the T_2 state to the singlet states.	The spin-orbit coupling (SOC) of the S_1 and T_1 states and of the S_1 and T_2 states of the materials were calculated. Energy transfer from the new host materials to the dopant was confirmed to occur via Förster resonance energy transfer (FRET) through the T_1 state to the singlet states.
3	Page 4, line 137	Although the ΔE_{ST} of S_1 – T_1 is relatively large (e.g., >0.2 eV), the T_2 state and the S_1 energy gap are relatively small, which can be interpreted as promoting RISC from the T_2 state to the S_1 level. This result means that the new host materials may promote activation of the RISC process because of the combination of the contribution of the T_2 state and the TADF properties.	Although ΔE_{ST} is smaller for S_1 – T_2 than for S_1 – T_1 (Supplementary Table S2), the RISC process occurs only in the T_1 state because of rapid internal conversion from T_2 to T_1 .
4	Page 5, line 178	In a two-component system, the newly synthesized host materials may facilitate the FRET process to the dopant through RISC activation of host T_2 level	In a two-component system, the newly synthesized host materials may facilitate the FRET process to the dopant through RISC activation of the host T_1 level.
5	Page 3, line 115 (in 2 nd revision)	On the basis of the reports of Hatakeyama's group, the newly synthesized host materials are expected to exhibit TADF properties and to utilize	These sentences have been deleted.

	/ Page 4, line 118 (in 1 st revision file)	the T ₂ energy level ^{18,26-28} . In addition, most of the materials have larger SOC values of $\langle T_2 \rangle$ between the S ₁ and T ₂ levels compared with the SOC value of $\langle T_1 \rangle$ between the S ₁ and T ₁ levels (Supplementary Fig. S3–S9 and Table S3). Recently several important studies have examined systems with small SOC values ($< 1 \text{ cm}^{-1}$) ^{5,28,29} . The six materials examined in the present work showed higher SOC values than in these previous works. These results show that the newly synthesized TDBA-based host materials have the advantage of increased luminous efficiency through activation of RISC by simultaneously utilizing the T ₂ state as well as conventional TADF characteristics.	
--	---	--	--

Issue 2: No correlation between k_{RISC} and EQE

Answer: We thank the reviewer for this advice. As a result of carefully re-evaluating all related data, we found that the k_{RISC} value of TDBA-Ph cited in the original manuscript was incorrect because of incorrect fitting data, as the reviewer pointed out. Based on the experimental data, we cannot provide the related values because we cannot generate a fitting line for TDBA-Ph transient PL data. In addition, the k_{RISC} values for mTDBA-Si and mTDBA-2Si were incorrect in Supplementary Table S4. The values for these two materials originally had units including 'x10⁴' but the values were not converted to the scale of 'x10⁵' in Table S4. Thus, the k_{RISC} values for mTDBA-Si and mTDBA-2Si were changed from 5.6 to 0.56 and from 8.66 to 0.87, respectively. We sincerely apologize for the mistakes in the TRPL fitting process as well as the typographical errors. The TRPL graphs of the non-doped films are now presented in Fig. 1d. If the reviewer believes the data for each film needs to be presented separately, please let us know and we will amend it again.

Before text)

The delayed lifetimes of TDBA-Ph, mTDBA-Ph, mTDBA-2Ph, TDBA-Si, mTDBA-Si, and mTDBA-2Si were 0.91, 1.25, 2.33, 3.53, 6.65, and 6.49 μs , respectively (Table 1 and Supplementary Table S4).

Revised text) Page 4, line 140

The delayed lifetimes of mTDBA-Ph, mTDBA-2Ph, TDBA-Si, mTDBA-Si, and mTDBA-2Si were 1.25, 2.33, 3.53, 6.65, and 6.49 μs , respectively (Table 1 and Supplementary Table S4).

Before)

Supplementary Table S4 | Rate constant for TDBA based host materials (non-doped film) at room temperature.

	TDBA-Ph	mTDBA-Ph	mTDBA-2Ph	TDBA-Si	mTDBA-Si	mTDBA-2Si
Φ	0.62	0.65	0.63	0.66	0.61	0.75

Φ_F	0.538	0.589	0.603	0.644	0.592	0.731
Φ_{TADF}	0.082	0.061	0.027	0.016	0.018	0.019
τ (ns)	9.35	10.01	8.92	7.80	11.1	6.50
τ_{TADF} (μ s)	0.91	1.25	2.33	3.53	6.65	6.49
k_F ($\times 10^7$)	5.75	5.87	6.76	8.25	5.32	11.2
k_{IC} ($\times 10^7$)	3.53	3.16	3.97	4.25	3.40	3.75
k_{ISC} ($\times 10^7$)	10.4	9.27	4.85	3.14	2.63	3.92
Φ_{IC}	0.33	0.317	0.354	0.332	0.379	0.244
Φ_{ISC}	0.13	0.09	0.04	0.02	0.03	0.03
k_{TADF} ($\times 10^5$)	6.82	5.22	2.71	1.87	9.18	1.16
k_{RISC} ($\times 10^5$)	4.23	3.39	1.71	1.23	5.60	8.66

After)

Supplementary Table S4 | Rate constant for TDBA based host materials (non-doped film) at room temperature.

	TDBA-Ph	mTDBA-Ph	mTDBA-2Ph	TDBA-Si	mTDBA-Si	mTDBA-2Si
Φ	0.62	0.65	0.63	0.66	0.61	0.75
Φ_F	-	0.589	0.603	0.644	0.592	0.731
Φ_{TADF}	-	0.061	0.027	0.016	0.018	0.019
τ (ns)	9.35	10.01	8.92	7.80	11.1	6.50
τ_{TADF} (μ s)	-	1.25	2.33	3.53	6.65	6.49
k_F ($\times 10^7$)	-	5.87	6.76	8.25	5.32	11.2
k_{IC} ($\times 10^7$)	-	3.16	3.97	4.25	3.40	3.75
k_{ISC} ($\times 10^7$)	-	9.27	4.85	3.14	2.63	3.92
Φ_{IC}	-	0.317	0.354	0.332	0.379	0.244

Φ_{ISC}	-	0.09	0.04	0.02	0.03	0.03
$k_{TADF}(\times 10^5)$	-	5.22	2.71	1.87	9.18	1.16
$k_{RISC}(\times 10^5)$	-	3.39	1.71	1.23	0.56	0.87

Before)

Table 1 | Summary of the photophysical properties of the TDBA-based materials.

	Solution			Neat Film									
	λ_{ab}^a (nm)	λ_{em}^a (nm)	FWHM (nm)	λ_{ab}^a (nm)	λ_{em}^a (nm)	FWHM (nm)	PLQY (%)	$E_S / E_T / \Delta E_{ST}^b$ (eV)	τ_d (μs) _c	HOMO ^d (eV)	LUMO ^d (eV)	E_g^d (eV)	
TDBA	383	403	27	—	—	—	—	—/—/—	—	6.02	2.87	3.15	
TDBA-Ph	388	408	27	392	418	66	62	3.06/ 2.82/ 0.24	0.91	5.75	2.65	3.10	
mTDBA-Ph	392	421	34	398	434	75	65	2.99/ 2.78/ 0.21	1.25	5.68	2.64	3.04	
mTDBA-2Ph	400	433	37	406	450	70	63	2.91/ 2.71/ 0.20	2.33	5.64	2.67	2.97	
TDBA-Si	389	411	27	393	425	73	66	3.02/ 2.77/ 0.25	3.53	5.73	2.64	3.09	
mTDBA-Si	393	421	33	398	449	79	61	3.01/ 2.73/ 0.28	6.65	5.71	2.66	3.05	
mTDBA-2Si	401	432	35	405	447	75	75	2.96/ 2.72/ 0.24	6.49	5.65	2.69	2.96	

^a Maximum wavelength in UV-Vis absorption and photoluminescence spectra. ^b Singlet and triplet energies measured in the neat film state as an onset value ($\Delta E_{ST} = S_1 - T_1$). ^c Delayed lifetime calculated by PL decay for a vacuum-deposited neat film ^d HOMO value measured by UV photoelectron yield spectroscopy (AC-2); the LUMO value was calculated from the optical bandgap.

After)

Table 1 | Summary of the photophysical properties of the TDBA-based materials.

	Solution			Neat Film									
	λ_{ab}^a (nm)	λ_{em}^a (nm)	FWHM (nm)	λ_{ab}^a (nm)	λ_{em}^a (nm)	FWHM (nm)	PLQY (%)	$E_S / E_T / \Delta E_{ST}^b$ (eV)	τ_d (μs) _c	HOMO ^d (eV)	LUMO ^d (eV)	E_g^d (eV)	
TDBA	383	403	27	—	—	—	—	—/—/—	—	6.02	2.87	3.15	
TDBA-Ph	388	408	27	392	418	66	62	3.06/ 2.82/ 0.24	-	5.75	2.65	3.10	
mTDBA-Ph	392	421	34	398	434	75	65	2.99/ 2.78/ 0.21	1.25	5.68	2.64	3.04	
mTDBA-2Ph	400	433	37	406	450	70	63	2.91/ 2.71/ 0.20	2.33	5.64	2.67	2.97	
TDBA-Si	389	411	27	393	425	73	66	3.02/ 2.77/ 0.25	3.53	5.73	2.64	3.09	
mTDBA-Si	393	421	33	398	449	79	61	3.01/ 2.73/ 0.28	6.65	5.71	2.66	3.05	
mTDBA-2Si	401	432	35	405	447	75	75	2.96/ 2.72/ 0.24	6.49	5.65	2.69	2.96	

^a Maximum wavelength in UV-Vis absorption and photoluminescence spectra. ^b Singlet and triplet energies measured in the neat film state as an onset value ($\Delta E_{ST} = S_1 - T_1$). ^c Delayed lifetime calculated by PL decay for a vacuum-deposited neat film. ^d HOMO value measured by UV photoelectron yield spectroscopy (AC-2); the LUMO value was calculated from the optical bandgap.

Issue 3: Additional description

As pointed out by the reviewer, host materials with TADF characteristics have intrinsic properties that typically cause intersystem crossing (ISC) to be faster than reverse intersystem crossing (RISC). The molecular arrangement of these host materials in the solid film state is such that excitons generated within the host undergo various quenching processes, making it challenging to achieve 'hyperfluorescence'.

In our study, the introduction of tetraphenylsilyl (TPS) groups into these host materials causes RISC in the non-doped film state to be slower than that of the corresponding TPS-free host materials, leading to an unavoidable increase in triplet density. However, TPS-containing films doped with 2 wt% v-DABNA showed increased RISC. We believe that the presence of the bulky TPS moieties may minimize unwanted Dexter Energy Transfer and thus allow effective harvesting of the singlet excited state of v-DABNA through a rapid hyperfluorescence-assisted FRET process. This behavior is not typically observed in conventional host materials. A similar phenomenon was reported in a recent paper by Suh's group (*Adv. Funct. Mater.* 2023, 2213461). In that study, a large rotor group was introduced on the edge of v-DABNA-based molecules. This group blocked the approach of TADF sensitizers and limited Dexter Energy Transfer, thus optimizing hyperfluorescence. We will continuously strive to improve charge balance and adjust the chemical structure of the host to maximize EL performance. Therefore, in the revised manuscript we suggest the following reasons why host materials containing TPS have high EQEs.

Added text) Page 7, line 256

Additionally, the long intermolecular distances in host materials containing the bulky TPS group may prevent triplet-triplet annihilation (TTA) and/or singlet-triplet annihilation (STA), leading to increased EQE²⁹.

Ref 29 : *Adv. Funct. Mater.* 2023, 2213461

REVIEWER COMMENTS

Reviewer #3 (Remarks to the Author):

The manuscript has been improved according to my original concerns. However, I doubt accuracy of photophysical data in Table S4. Given Figure 1 and Table S5, it seems that the authors erroneously measured the TADF components due to the small proportion of the TADF components (0.019 to 0.082) and the mis-inclusion of the IRF components. It is necessary to show IRF in Figure 1 to prove the accuracy of their estimation of the rate constants. I assume that $\tau(\text{TADF})$ of non-doped film should be much longer than those estimated here, and thus $k(\text{ISC})$ and $k(\text{RISC})$ should be much smaller. Since their TADF properties have not accurately estimated, at this stage I do not agree with Figure 3a that suggests rapid RISC in TDBA.

REVIEWER COMMENTS

Reviewer #3 (Remarks to the Author):

The manuscript has been improved according to my original concerns. However, I doubt accuracy of photophysical data in Table S4. Given Figure 1 and Table S5, it seems that the authors erroneously measured the TADF components due to the small proportion of the TADF components (0.019 to 0.082) and the mis-inclusion of the IRF components. It is necessary to show IRF in Figure 1 to prove the accuracy of their estimation of the rate constants. I assume that $\tau(\text{TADF})$ of non-doped film should be much longer than those estimated here, and thus $k(\text{ISC})$ and $k(\text{RISC})$ should be much smaller. Since their TADF properties have not accurately estimated, at this stage I do not agree with Figure 3a that suggests rapid RISC in TDBA.

Issue 1: The manuscript has been improved according to my original concerns. However, I doubt accuracy of photophysical data in Table S4. Given Figure 1 and Table S5, it seems that the authors erroneously measured the TADF components due to the small proportion of the TADF components (0.019 to 0.082) and the mis-inclusion of the IRF components.

Answer: We thank the reviewer for these important comments. In response to the reviewer's concern, we rechecked the instrument and the related measurement program to confirm the related data for the non-doped film state in Table S4. We found no error in the data in Table S4. All the related data, including the fitting line, were measured and calculated by the equipment program as mentioned in the text below (Page 9, line 309 in the Methods part of main manuscript).

Page 9, line 309

“Photoluminescent decay traces were obtained through time-correlated single-photon coefficient (TCSPC) technology using PicoQuant, FluoTime 250 instruments (PicoQuant, Germany). A 377 nm pulse laser was used as an excitation source and data analysis was performed using the exponential fitting model of the FluoFit software.”

To address the reviewer's concern, we will add the following sentence if the reviewer recommends it:

Added text) Page 4, line 141

In addition, the quantum yield values of the TADF portion of the non-doped film state were in the range of 0.018 to 0.061, which is relatively low compared to the high-performance TADF materials.

Issue 2: It is necessary to show IRF in Figure 1 to prove the accuracy of their estimation of the rate constants.

Answer: We apologize for omitting the IRF data curve. The relevant data was added to Fig. 1d as follows:

Before)

Fig. 1 | **a** Chemical structure of the newly synthesized TDDBA-based host materials. **b** Photoluminescence spectra of the TDDBA-based host materials in toluene (0.01 mM). **c** Photoluminescence spectra of the TDDBA-based host materials as neat films. **d** Transient photoluminescence decay spectra of the neat films.

After)

Fig. 1 | **a** Chemical structure of the newly synthesized TDDBA-based host materials. **b** Photoluminescence spectra of the TDDBA-based host materials in toluene (0.01 mM). **c**

Photoluminescence spectra of the TDBA-based host materials as neat films. **d** Transient photoluminescence decay spectra of the neat films (IRF: instruments response function).

Issue 3: I assume that tau (TADF) of non-doped film should be much longer than those estimated here, and thus k(ISC) and k(RISC) should be much smaller. Since their TADF properties have not accurately estimated, at this stage I do not agree with Figure 3a that suggests rapid RISC in TDBA.

Answer: We appreciate the reviewer's concern. However, these data were obtained from the fitting curve automatically provided by the equipment software. These values in Table S4 were provided by the PICO company program, as mentioned above.

Also, a system with a similar data scale (i.e., short τ_{TADF} and large k(RISC)) was described in Prof. Kwon's recent paper (Angew. Chem. Int. Ed., e202306768, 2023).

Emitter	Φ_{PL}	Φ_p	Φ_d	τ_p [ns]	τ_d [μ s]	k_p [10^8 /s]	k_d [10^5 /s]	k_r^s [10^7 /s]	k_{ISC} [10^7 /s]	k_{RISC} [10^5 /s]
NO-DBMR	0.834	0.724	0.110	6.64	5.46	1.51	1.83	10.9	4.16	1.01
Cz-DBMR	0.821	0.657	0.164	7.54	1.96	1.33	5.10	8.71	4.55	3.72

In the present paper, we wish to focus only on the material structure affording highly efficient blue EL. We will prepare another manuscript that includes the excitation dynamics and the energy transfer mechanism based on transient absorption spectroscopy, as well as other experiments. Please understand this situation.

Based on the reviewer's comments, we toned down the sentences as follows.

Revised text)

No.	Position	Before	After
1	Page 7, line 233	The high efficiency and low roll-off might be due to the prevention of non-radiative processes, which occurs via a fast energy transfer process from the new TDBA-Si host to the ν -DABNA dopant.	The high efficiency and low roll-off might be due to the prevention of non-radiative processes.
2	Page 7, line 241	First, the enhanced EL performance can be explained by the rapid energy transfer from the TPS-based host materials to the dopant.	First, the enhanced EL performance can be explained by the energy transfer from the TPS-based host materials to the dopant.